# Training-Free Robust Multimodal Learning via Sample-Wise Jacobian Regularization

## Abstract

Multimodal fusion emerges as an appealing technique to improve model performances on many tasks. Nevertheless, the robustness of such fusion methods is rarely involved in the present literature. In this paper, we are the first to propose a training-free robust late-fusion method by exploiting conditional independence assumption and Jacobian regularization. Our key is to minimize the Frobenius norm of a Jacobian matrix, where the resulting optimization problem is relaxed to a tractable Sylvester equation. Furthermore, we provide a theoretical error bound of our method and some insights about the function of the extra modality. Several numerical experiments on AV-MNIST, RAVDESS, and VGGsound demonstrate the efficacy of our method under both adversarial attacks and random corruptions.

## 1 Introduction

Deep fusion models have recently drawn great attention of researchers in the context of *multimodal learning* (Vielzeuf et al., 2018; Baltrušaitis et al., 2018; Pérez-Rúa et al., 2019; Wang et al., 2020; Xue et al., 2021) as it provides an easy way to increase model accuracy and robustness. For instance, RGB cameras and LiDARs are usually deployed simultaneously on an autonomous vehicle, and the resulting RGB images and point clouds are referred to as two modalities, respectively. When RGB images are blurry at night, point clouds could provide complementary information and help to make decisions in vision tasks (Kim & Ghosh, 2019). Over the past few years, numerous multimodal fusion methods have been proposed at different levels: early-, middle-, and late-fusion (Chen et al., 2021). In early-fusion, input feature vectors from different modalities are concatenated and fed into one single deep neural network (DNN), while in middle-fusion, they go into DNNs independently and exchange information in feature space. Unlike the previous two cases, late-fusion is realized by merging distinct DNNs at their output layers via concatenation, element-wise summation, etc.

These three levels of fusion possess different pros and cons. For instance, late-fusion, the primary concern of our paper, is (i) privacy-friendly and (ii) convenient to deploy. Specifically, assume that a hospital wants to have an AI agent to judge whether a patient has a certain disease or not (Sun et al., 2020). It has to divide the complete training feature (e.g., medical records, X-ray images) of every patient and deliver them to different AI companies, otherwise, the patients' identities will be exposed and their privacy are unprotected. This, in turn, directly rules out the possibility of applying early- or middle-fusion methods. On the other hand, the hospital could still exploit late-fusion technique to generate the ultimate AI agent after several unimodal DNNs are trained by AI companies. Moreover, unlike early- or middle-fusion, many late-fusion methods could tolerate missing modality information (i.e., no need for paired data) and thus are convenient to deploy.

Although late-fusion is a mature topic in the literature, its performance under adversarial attacks (Madry et al., 2018; Tsipras et al., 2019) and random corruptions (Zheng et al., 2016; Kim & Ghosh, 2019) is rather under-explored. In this paper, we address the problem of robust late-fusion by utilizing Jacobian regularization (Varga et al., 2017; Jakubovitz & Giryes, 2018; Hoffman et al., 2019; Chan et al., 2019) and conditional independence assumption (Sun et al., 2020). The key is to minimize the Frobenius norm of a Jacobian matrix so that the multimodal prediction is stabilized (see Figure 1). Our main contributions are as follows:

- To the best of our knowledge, we are the first to propose a training-free robust late-fusion method. The involving optimization problem is relaxed to a Sylvester equation (Jameson, 1968), and the solution is obtained with only a little computational overhead.

- We provide a theoretical error bound of our proposed robust late-fusion method and an illustrative explanation about the function of the extra modality via the TwoMoon example.
- Thorough numerical experiments demonstrate that our method outperforms other late-fusion methods and is capable to handle both adversarial attacks and random corruptions.

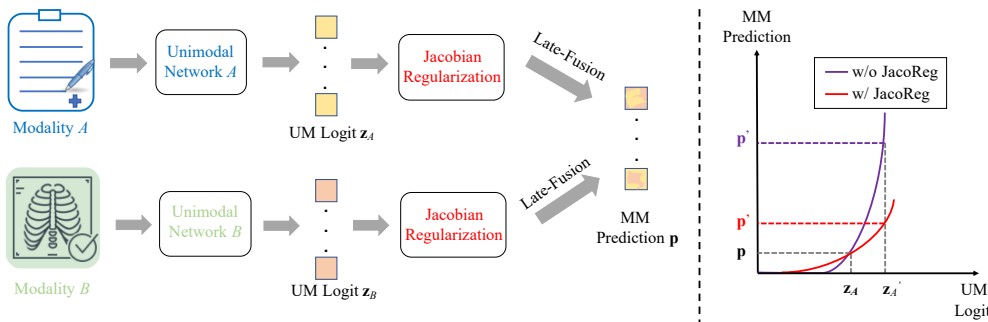

Figure 1: Illustration of the proposed robust late-fusion method. Before unimodal raw logit $\mathbf{z}_A$ and $\mathbf{z}_B$ are fused, the Jacobian regularization technique is applied. Roughly speaking, it will enforce the derivative of $\mathbf{p}$ with respect to $\mathbf{z}_A$ becomes smaller. Thus, when $\mathbf{z}_A$ is perturbed to $\mathbf{z}'_A$ (due to random corruption or adversarial attack), the change of multimodal prediction $||\mathbf{p}' - \mathbf{p}||$ will be limited to some extent. For the illustration purpose, all variables here (e.g., $\mathbf{p}, \mathbf{z}_A$) are drawn in one-dimensional space.

## 2 PRELIMINARY

**Network Robustness**   To verify the network robustness, two major kinds of perturbations are used, which in our paper we referred to as (i) adversarial attacks such as FGSM, PGD, or CW attack (Goodfellow et al., 2014; Madry et al., 2018; Carlini & Wagner, 2017) and (ii) random corruptions such as Gaussian noise, missing entries or illumination change (Zheng et al., 2016; Kim & Ghosh, 2019). Correspondingly, many methods have been proposed to offset the negative effects of such perturbations. Adversarial training based on projected gradient descent (Madry et al., 2018) is one strong mechanism to defend against adversarial attacks, and the recent Free-m method (Shafahi et al., 2019) is proposed as its fast variant. Besides adversarial training, several regularization techniques are also proven to have such capability, such as Mixup (Zhang et al., 2020), Jacobian regularization (Jakubovitz & Giryes, 2018). Alternatively, with regard to random corruptions, Mixup and Jacobian regularization are also effective in this case (Zhang et al., 2018; Hoffman et al., 2019). Another powerful approach is stability training (Zheng et al., 2016), where it introduces an additional KL divergence term into the conventional classification loss so that the trained DNNs are stabilized.

**Multimodal Learning**   DNNs trained by fusing data from different modalities have outperformed their unimodal counterparts in various applications, such as object detection (Chen et al., 2021; Kim & Ghosh, 2019), semantic segmentation (Chen et al., 2020b; Feng et al., 2020), audio recognition (Gemmeke et al., 2017; Chen et al., 2020a). Based on where the information is exchanged between different modalities, multimodal fusion methods could be classified into three kinds: (i) early-fusion (Wagner et al., 2016; Chen et al., 2021), (ii) middle-fusion (Kim & Ghosh, 2019; Wang et al., 2020), and (iii) late-fusion (Chen et al., 2021; Liu et al., 2021). For instance, if the information is fused at the end of DNNs (e.g., Figure 1), such a method belongs to the late-fusion category.

Although vast efforts have been put into exploiting multimodal fusion methods to improve DNNs' accuracy on specific learning tasks, few works have explored network robustness in the multimodal context. Specifically, Mees et al. (2016), Valada et al. (2017), and Kim et al. (2018) exploited gating networks to deal with random corruptions, adverse or changing environments. Afterwards, Kim & Ghosh (2019) proposed a surrogate minimization scheme and a latent ensemble layer to handle single-source corruptions. However, all the aforementioned methods belong to middle-fusion and only random corruptions are considered. On the other hand, we focus on another important scenario: late-fusion, and besides corruptions, we further take adversarial attacks into account. To the best of our knowledge, robust late-fusion is un-explored in the previous literature.

## 3 Jacobian Regularization in Test Time

Consider a supervised $K$-class classification problem in the multimodal context. Suppose that features from two distinct modalities $A$ (e.g., audio) and $B$ (e.g., video) are provided in the form of $\mathcal{D}_A = \{(\mathbf{x}_A^i, \mathbf{y}^i)\}_{i=1}^N$ and $\mathcal{D}_B = \{(\mathbf{x}_B^i, \mathbf{y}^i)\}_{i=1}^N$, where $\{\mathbf{x}_A, \mathbf{x}_B\}$ represents input features and $\mathbf{y}$ represents the true label. We train two unimodal networks separately, each corresponding to one modality. Given a specific input feature $\mathbf{x}_A$, the first unimodal network calculate the class prediction $\mathbf{p}_A \in \mathbb{R}^K$ by:

$$\mathbf{p}_A = \boldsymbol{\sigma}_1(\mathbf{z}_A) = \boldsymbol{\sigma}_1(\mathbf{W}_A \mathbf{h}_A + \mathbf{b}_A) = \boldsymbol{\sigma}_1(\mathbf{W}_A \mathbf{f}_A(\mathbf{x}_A) + \mathbf{b}_A) \tag{1}$$

where $\boldsymbol{\sigma}_1(\cdot)$ represents the Softmax function, $\mathbf{z}_A \in \mathbb{R}^K$ represents the raw logit, and $\mathbf{h}_A = \mathbf{f}_A(\mathbf{x}_A) \in \mathbb{R}^H$ represents the feature being fed into the last layer [1]. Here $\mathbf{W}_A \in \mathbb{R}^{K \times H}$ and $\mathbf{b}_A \in \mathbb{R}^K$ are the learnable weight and bias of the last linear layer, respectively. Similarly, the second unimodal network provides the class prediction $\mathbf{p}_B = \boldsymbol{\sigma}_1(\mathbf{z}_B) = \boldsymbol{\sigma}_1(\mathbf{W}_B \mathbf{h}_B + \mathbf{b}_B) = \boldsymbol{\sigma}_1(\mathbf{W}_B \mathbf{f}_B(\mathbf{x}_B) + \mathbf{b}_B)$.

Based on the conditional independence assumption (Kong & Schoenebeck, 2018), the basic statistical late-fusion method generates the final class prediction as (Chen et al., 2021):

$$\mathbf{p} = \boldsymbol{\sigma}_2\left(\frac{\mathbf{p}_A \odot \mathbf{p}_B}{\mathbf{freq}}\right) = \boldsymbol{\sigma}_2\left(\frac{\boldsymbol{\sigma}_1(\mathbf{z}_A) \odot \boldsymbol{\sigma}_1(\mathbf{z}_B)}{\mathbf{freq}}\right) \tag{2}$$

where the division is performed in an element-wise manner, $\odot$ represents the element-wise product, and $\boldsymbol{\sigma}_2(\cdot)$ represents a linear normalization enforcing the summation of elements equal to one. Here $\mathbf{freq} \in \mathbb{R}^K$ contains the occurring frequencies of each class, calculated from the training dataset.

Our proposed approach builds upon (2). Specifically, we consider adding two $K \times K$ weight matrices $\{\mathbf{W}_a, \mathbf{W}_b\}$ ahead of $\{\mathbf{z}_A, \mathbf{z}_B\}$, respectively, right before they get activated. Consequently, the final multimodal prediction is re-calibrated as:

$$\mathbf{p}' = \boldsymbol{\sigma}_2\left(\frac{\boldsymbol{\sigma}_1(\mathbf{z}_A') \odot \boldsymbol{\sigma}_1(\mathbf{z}_B')}{\mathbf{freq}}\right) = \boldsymbol{\sigma}_2\left(\frac{\boldsymbol{\sigma}_1(\mathbf{W}_a \mathbf{z}_A) \odot \boldsymbol{\sigma}_1(\mathbf{W}_b \mathbf{z}_B)}{\mathbf{freq}}\right) \tag{3}$$

Suppose that the data provided from modality $A$ are perturbed at the input or feature level, while the data from modality $B$ are clean. For matrix $\mathbf{W}_b$, we could simply set it to an identity matrix, implying that we didn't invoke the robust add-on for the second modality. To determine the value of $\mathbf{W}_a$, we first calculate the derivative of $\mathbf{p}'$ with respect to $\mathbf{h}_A$ (see Appendix A.1):

$$\frac{\partial \mathbf{p}'}{\partial \mathbf{h}_A} = \mathbf{J}' \mathbf{W}_a \mathbf{W}_A = [\mathbf{p}' \mathbf{p}'^{,T} - \mathrm{Diag}(\mathbf{p}')] \mathbf{W}_a \mathbf{W}_A \tag{4}$$

Then we minimize the following regularized Jacobian loss with respect to $\mathbf{W}_a$:

$$\min_{\mathbf{W}_a} L = \min_{\mathbf{W}_a} (1 - \gamma) ||\mathbf{J}' \mathbf{W}_a \mathbf{W}_A||_F^2 + \gamma ||\mathbf{W}_a - \mathbf{I}||_F^2 \tag{5}$$

where $0 < \gamma < 1$ is a tunable hyper-parameter. Minimizing the first term in the loss could make the change of $\mathbf{p}'$ limited to some extent when $\mathbf{h}_A$ is perturbed, while the second term in the loss guarantees numerical stability and the prediction in the perturbed case won't get too far from that of the clean case. For a specific multimodal input $\{\mathbf{x}_A, \mathbf{x}_B\}$, once $\mathbf{W}_a$ is determined, so are $\mathbf{p}'$ and $\mathbf{J}'$ via (3) and (4), respectively. Thus, (5) is well-determined and non-linear with respect to $\mathbf{W}_a$.

We propose a heuristic iterative method making the above optimization problem tractable in Algorithm 1. Our key is to decouple $\mathbf{J}'$ from $\mathbf{W}_a$. Namely, in step 5 of Algorithm 1, all terms are known except $\mathbf{W}_a$, and thus the relaxed loss is convex. After writing out all terms of $\partial L^{(t)} / \partial \mathbf{W}_a$, we observe that it is a Sylvester equation (Jameson, 1968). It has a unique solution and the run time is as large as inverting a $K \times K$ matrix and hence affordable. See Appendix A.2 for details.

**Remarks** First, in our implementation we find that one iteration could already yield a sufficiently accurate result, thus all our numerical results are reported with $t_{\max} = 1$ (i.e., $\mathbf{p}' = \mathbf{p}^{(1)}$). Second, if we know data from modality $B$ are also perturbed, we could solve $\mathbf{W}_b$ in a similar manner. Third,

---

[1](1) holds because the last layer is usually implemented as a fully connected (FC) layer with Softmax activation in classification tasks.

---

**Algorithm 1** Iteratively solving regularized Jacobian loss

---

1: Given one specific input $\{\mathbf{x}_A, \mathbf{x}_B\}$. Initialize iteration index $t = 0$.
2: Perform one forward pass, yielding the initial class prediction $\mathbf{p}^{(0)}$.
3: **while** $t < t_{\max}$ **do**
4:     Calculate $\mathbf{J}^{(t)} = \mathbf{p}^{(t)}\mathbf{p}^{(t),T} - \text{Diag}(\mathbf{p}^{(t)})$.
5:     Minimize $L^{(t)} = (1 - \gamma)||\mathbf{J}^{(t)}\mathbf{W}_a\mathbf{W}_A||_F^2 + \gamma||\mathbf{W}_a - \mathbf{I}||_F^2$ with respect to $\mathbf{W}_a$.
6:     Calculate $\mathbf{p}^{(t+1)}$ based on Eq (3) with the optimal $\mathbf{W}_a$.
7:     Update $t = t + 1$.
8: **end while**
9: Return $\mathbf{p}' = \mathbf{p}^{(t)}$.

---

notice that our approach is invoked merely during inference (i.e., test time), but not the train time. Furthermore, we demonstrate our approach in the context of two modalities, while obviously, it could be equally applied to many modalities. Finally, with a moderate assumption, we can further prove that when the input is perturbed, the change of final prediction enhanced by our method is limited to a certain amount. This has been summarized in Theorem 1 (see Appendix A.3). Some immediate corollary include: (i) when $\boldsymbol{\epsilon} \sim N(\mathbf{0}, \boldsymbol{\Sigma})$, the bound is simplified to $E[||\mathbf{p}'^{,noise} - \mathbf{p}'||] \leq l\left(\frac{\gamma K}{2(1-\gamma)}\right)^{1/2} \text{Tr}[\boldsymbol{\Sigma}]$, (ii) when the $L_2$ norm of $\boldsymbol{\epsilon}$ is constrained smaller than $\delta$ (usually assumed in adversarial attacks), the bound is simplified to $||\mathbf{p}'^{,noise} - \mathbf{p}'|| \leq l\,\delta\left(\frac{\gamma K}{2(1-\gamma)}\right)^{1/2}$.

**Theorem 1** *If $\mathbf{f}_A$ is $l$-Lipschitz continuous and $\mathbf{x}_A$ is perturbed by $\boldsymbol{\epsilon}$: $\mathbf{x}_A^{noise} = \mathbf{x}_A + \boldsymbol{\epsilon}$, then the Euclidean norm of our final prediction (i.e., $||\mathbf{p}'^{,noise} - \mathbf{p}'||$) at most changes $l\sqrt{\frac{\gamma K}{2(1-\gamma)}}||\boldsymbol{\epsilon}||$.*

## 4 NECESSITY OF THE EXTRA MODALITY: BIASING EFFECT

In this sub-section, we explain the principles behind our approach. Let us move one step back and consider the unimodal case. In what follows, we will demonstrate that our approach might not work well in the unimodal context, which in turn justifies the necessity of the extra modality.

At first glance, our approach seems to be equally applicable in the unimodal context. Namely, suppose that only one unimodal network $\mathbf{p} = \boldsymbol{\sigma}_1(\mathbf{z}) = \boldsymbol{\sigma}_1(\mathbf{W}\mathbf{h} + \mathbf{b})$ is available, in test time and for each specific input $\mathbf{x}$, we add a weight matrix $\mathbf{W}_x$ before the raw logit $\mathbf{z}$ being fed into the Softmax activation, re-calibrating the final unimodal prediction as: $\mathbf{p}' = \boldsymbol{\sigma}_1(\mathbf{z}') = \boldsymbol{\sigma}_1(\mathbf{W}_x\mathbf{z})$. Here $\mathbf{W}_x$ can be solved by analogy to (5). However, we observe that the introduction of $\mathbf{W}_x$ usually won't change the final prediction.

**TwoMoon example** For an intuitive understanding, we consider the TwoMoon example used by Xue et al. (2021). In this example, all data points are scattered in a 2D space. The data located at the upper and lower leaf have true labels 0 and 1, and are colored by red and blue, respectively.

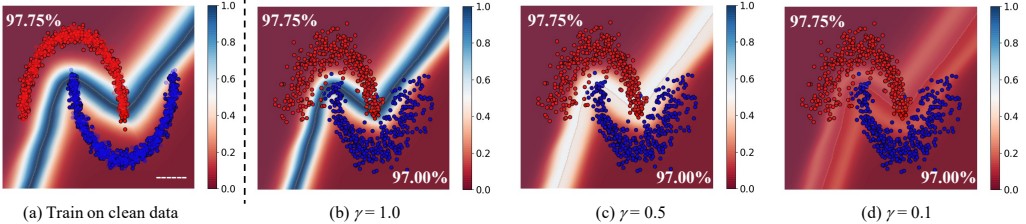

Figure 2: (Unimodal case) The leftmost figure reveals the results of training and testing on clean data. The heatmap in the background represents the value of $||\mathbf{J}\mathbf{W}||_F$. The remaining right three figures show the results of testing on data with Gaussian noise, and similarly, the heatmap in the background represents the value of $||\mathbf{J}\mathbf{W}_x\mathbf{W}||_F$. For each figure, the test accuracy on clean and noisy data are reported at the left top and right bottom.

In the unimodal case, we take both horizontal and vertical coordinates as input and train a neural network with three FC layers. As shown in Figure 2 (a), the network perfectly fits to the clean data and achieves 97.75% accuracy on the clean test data. In the remaining three figures, we evaluate the trained network on noisy test data. Specifically, we deliberately choose $\gamma = 1.0$ in Figure 2 (b), so that our approach is actually not invoked (since the solved $\mathbf{W}_x$ equals $\mathbf{I}$). In this case, the accuracy drops to 97.00% on noisy data, while the heatmap in the background doesn't change at all compared to Figure 2 (a). In Figure 2 (c) and (d), we choose $\gamma = 0.5$ and $\gamma = 0.1$, respectively. We observe that even though the color of the heatmap surely becomes lighter as expected, the test accuracies of both cases are still 97.00%. More importantly, we find that this phenomenon is not coincident and that the prediction doesn't change for any input after adding $\mathbf{W}_x$ in unimodal binary classification. See Appendix A.4 for rigorous mathematical proof. Moreover, in a $K$-class ($K > 2$) classification problem, the final prediction might change if $\gamma$ is sufficiently small. For instance, given $\mathbf{z} = [1, 0, 2]^T$ and $\mathbf{W} = \mathbf{I}$, if we choose $\gamma = 0.5$, then the final prediction will change from $\mathbf{p} = [0.245, 0.090, 0.665]^T$ to $\mathbf{p}' = [0.270, 0.096, 0.635]^T$, where the last entry remains to be the largest. However, if we choose $\gamma = 0.01$, then the final prediction will become $\mathbf{p}' = [0.391, 0.219, 0.390]^T$, and now the first entry is the largest. See Appendix A.5 for a theoretical bound of $\gamma$ in this high-dimensional case.

Now we turn to the multimodal case. We treat the horizontal coordinates as one modality, and the vertical coordinates as the second modality. Two neural networks are trained, each corresponding to one modality. Then we fuse them based on the aforementioned statistical fusion method. As shown in Figure 3, in the multimodal context, when our method is invoked ($\gamma = 0.5$ or $0.1$), the color of the heatmap becomes lighter and the test accuracies on noisy data all increase compared to the trivial statistical fusion (i.e., when $\gamma = 1.0$). On the other hand, the test accuracies on clean data almost remain still (or slightly increase alongside $\gamma \downarrow$).

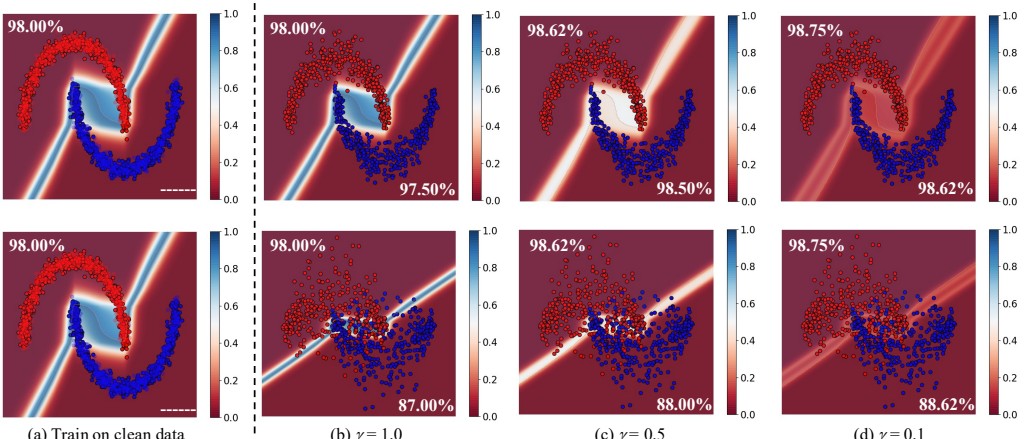

Figure 3: (Multimodal case) The leftmost column reveals the results of training and testing on clean data. For illustration purpose, we only perturb the Y-coordinates with Gaussian noise. The first and second row correspond to small and large noise, respectively. Heatmaps reveal the values of $||\mathbf{J}\mathbf{W}||_F$ in (a) and $||\mathbf{J}\mathbf{W}_A\mathbf{W}||_F$ in (b), (c), and (d). For each figure, the test accuracy on clean and noisy data are reported at the left top and right bottom.

**Lemma 1** *Statistical fusion is equivalent to concatenating the raw logits:*

$$\sigma_2\left(\frac{\mathbf{p}_A \odot \mathbf{p}_B}{\mathbf{freq}}\right) = \sigma_1(\mathbf{z}_A + \mathbf{z}_B - \ln\mathbf{freq}) = \sigma_1\left(\begin{bmatrix} \mathbf{I} & \mathbf{I} \end{bmatrix}\begin{bmatrix} \mathbf{z}_A \\ \mathbf{z}_B \end{bmatrix} - \ln\mathbf{freq}\right)$$

**Biasing effect of the extra modality** As we have seen in the TwoMoon example, our proposed method behaves differently in the unimodal and multimodal context. To get a better understanding of this subtle difference, we first summarize an equivalent form of statistical fusion in Lemma 1 (see Appendix A.1). Now assume that the sample-wise $\mathbf{W}_a$ is added, then the final multimodal prediction becomes $\sigma_1(\mathbf{W}_a\mathbf{z}_A + \mathbf{z}_B - \ln\mathbf{freq})$. Alternatively, in the unimodal context, our method adds a sample-wise $\mathbf{W}_x$ and the final prediction becomes $\sigma_1(\mathbf{W}_x\mathbf{z})$. Comparing these two expressions,

we observe that the introduced $\mathbf{W}_a$ and $\mathbf{W}_x$ occur in the perturbed modality. However, in the multimodal case, the extra modality acts as a bias term inside the Softmax function $\boldsymbol{\sigma}_1$ and plays a key role.

# 5 EXPERIMENTAL RESULTS

## 5.1 AV-MNIST

AV-MNIST (Pérez-Rúa et al., 2019; Vielzeuf et al., 2018) is a novel dataset created by pairing audio features to the original MNIST images. The first modality corresponds to MNIST images with size of $28 \times 28$ and $75\%$ energy removed by principal component analysis (Pérez-Rúa et al., 2019). The second modality is made up of spectrograms with size of $112 \times 112$. These spectrograms are extracted from audio samples obtain by merging pronounced digits and random natural noise. Following Vielzeuf et al. (2018), we use LeNet5 (LeCun et al., 1989) and a 6-layer CNN to process image and audio modalities, respectively. To thoroughly verify the efficiency of our method, we design various types of perturbations: (i) random corruptions including Gaussian noise ($\omega_0$), missing entries ($\omega_1$), and biased noise ($\omega_2, \omega_3$), and (ii) adversarial attacks including FGSM ($\omega_4$) and PGD attack ($\omega_5, \omega_6$). See Appendix B.1 for detailed definitions.

**Baselines** We limit our comparison to the range of late-fusion methods. To verify our method, we consider several methods which could improve network robustness, including (i) regular training, (ii) adversarial training (advT) by Madry et al. (2018), (iii) free-m training (freT) by Shafahi et al. (2019), (iv) stability training (stabT) by Zheng et al. (2016), and (v) Mixup (mixT) by Zhang et al. (2018). To the best of our knowledge, no previous robust late-fusion methods exist. Thus for comparison purpose, we slightly modify the confidence-based weighted summation in Yu et al. (2021) and adopt it here, referred to as mean fusion with confidence-based weighted summation. Combinations of different fusion methods and robust add-ons provide us with a few baselines. We emphasize that in our experiments, the train data are always free of noise. Following Kim & Ghosh (2019), our experiments are mainly conducted on the case when one modality is perturbed.

**Results** Table 1 reports the test accuracies of several models on AV-MNIST. Six unimodal networks are trained on clean data. Then we fuse them with different fusion methods and different robust add-ons. We only invoke our Jacobian regularization for the perturbed audio modality, while keeping the image modality unchanged. First, in the case of mean fusion, confidence-based weighted summation doesn't improve robustness. In the case of statistical fusion, we observe that our proposed Jacobian regularization method not only boosts the accuracy on noisy data but also on clean data. For instance, when our method is invoked on the combination of model-0 and model-1, the test accuracy on clean data increases from $93.6\%$ to $94.7\%$, and the test accuracy on data with Gaussian noise (e.g., $\omega_0 = 1.0$) increases from $80.4\%$ to $84.5\%$. Similar phenomena can be observed when our method is invoked on other combinations. Moreover, the number of green arrows is much larger than that of red arrows implying that our method is compatible with different robust techniques and applicable under different noises/attacks.

Comparing the last row in the second sub-section and the third row in the third sub-section, we find that with our proposed method enabled, even if the unimodal backbone model-1 is worse than model-2, the final multimodal accuracies $(0, 1)$ with our method invoked are even larger than $(0, 2)$ without our method invoked. Similar phenomena could usually be observed in other sub-sections. We argue that in this example, statistically fusing two regular unimodal baselines with our Jacobian regularization invoked surpasses robust unimodal baselines and their trivial fusion. Moreover, the highest accuracy in each column has been bolded. It is clear that the multimodal network with our method invoked usually achieves the best performance over all others.

Figure 4 further plots model accuracy versus magnitude of noise/attack. It displays a consistent trend that our method works well regardless of the magnitude of noise/attack. Furthermore, the larger the noise/attack is, the better our method performs. More numerical results (such as the impact of hyper-parameters, applying perturbation on the image modality) are shown in Appendix B.2.

Table 1: Accuracies of different models (%) are evaluated on AV-MNIST when audio features are perturbed. Mean accuracies are reported after repeatedly running 20 times. The value of $\gamma$ is selected from $\{0.1, 0.5, 0.9\}$. The green (↑) or red (↓) arrow represents after applying our Jacobian regularization, the model accuracy increases or decreases compared to others with the same unimodal backbones. The best accuracy in one column is bolded. 'UM' and 'MM' represent 'unimodal' and 'multimodal', respectively. 'MM$(0, i)$' represents a multimodal network obtained by fusing the unimodal network indexed by '0' and '$i$'.

| UM / MM | Model | Clean | $\omega_0 = 1.0$ | $\omega_1 = 10$ | $\omega_4 = 0.03$ | $\omega_5 = 0.001$ |
|---|---|---|---|---|---|---|
| UM Nets | 0: Img-regT | 73.4 | 73.4 | 73.4 | 73.4 | 73.4 |
| | 1: Aud-regT | 83.9 | 55.1 | 73.3 | 69.9 | 77.8 |
| | 2: Aud-advT | 84.4 | 59.2 | 72.0 | 81.9 | 83.3 |
| | 3: Aud-freT | 82.1 | 55.6 | 71.9 | 80.8 | 81.6 |
| | 4: Aud-staT | 86.2 | 67.6 | 74.4 | 66.5 | 74.5 |
| | 5: Aud-mixT | 87.6 | 61.3 | 74.9 | 74.9 | 78.3 |
| MM (0, 1) | Mean-w/o | 93.6 | 80.3 | 86.4 | 89.8 | 92.7 |
| | Mean-w/ | 90.5 | 73.7 | 83.6 | 82.0 | 88.2 |
| | Stat-w/o | 93.6 | 80.4 | 86.6 | 89.9 | 92.7 |
| | Stat-w/ (ours) | 94.7 (↑) | 84.5 (↑) | 89.1 (↑) | 92.2 (↑) | **94.1** (↑) |
| MM (0, 2) | Mean-w/o | 91.8 | 78.3 | 82.7 | 91.9 | 92.5 |
| | Mean-w/ | 86.0 | 72.7 | 77.5 | 85.7 | 86.9 |
| | Stat-w/o | 91.7 | 78.3 | 83.5 | 91.9 | 92.4 |
| | Stat-w/ (ours) | 93.4 (↑) | 83.1 (↑) | 86.4 (↑) | **93.5** (↑) | 93.7 (↑) |
| MM (0, 3) | Mean-w/o | 93.0 | 81.1 | 87.7 | 93.2 | 93.2 |
| | Mean-w/ | 82.4 | 74.4 | 78.6 | 83.7 | 83.5 |
| | Stat-w/o | 93.0 | 80.9 | 87.7 | 93.2 | 93.2 |
| | Stat-w/ | 93.0 (↑) | 82.3 (↑) | 88.1 (↑) | 92.7 (↓) | 92.9 (↓) |
| MM (0, 4) | Mean-w/o | 93.0 | 83.8 | 84.9 | 83.5 | 88.8 |
| | Mean-w/ | 90.9 | 78.1 | 81.8 | 74.6 | 81.3 |
| | Stat-w/o | 93.1 | 83.7 | 85.3 | 83.4 | 88.8 |
| | Stat-w/ (ours) | 94.7 (↑) | **87.5** (↑) | 87.8 (↑) | 87.9 (↑) | 91.9 (↑) |
| MM (0, 5) | Mean-w/o | **95.2** | 85.9 | 89.7 | 91.1 | 92.5 |
| | Mean-w/ | 94.0 | 82.1 | 88.6 | 87.1 | 88.9 |
| | Stat-w/o | 95.1 | 85.7 | **90.1** | 91.1 | 92.4 |
| | Stat-w/ (ours) | 95.0 (↓) | 86.1 (↑) | 90.1 (↓) | 91.0 (↓) | 92.2 (↓) |

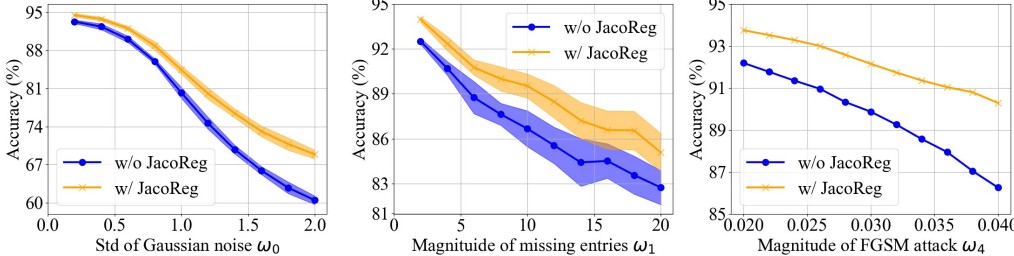

Figure 4: Accuracies of multimodal networks obtained by statistically fusing a vanilla image network and audio network with $\gamma = 0.1$. Mean and Std of 20 repeated experiments are shown by the solid lines and the shaded regions, respectively. Note that FGSM attack is rather deterministic, thus there is almost no variance when repeating experiments.

## 5.2 EMOTION RECOGNITION ON RAVDESS

We consider emotion recognition on RAVDESS (Livingstone & Russo, 2018; Xue et al., 2021). We use a similar network structure for both the image and audio modality: Three convolution layers, two max pooling layers, and three FC layers connect in sequential. Similar baselines and experimental settings are adopted as those in AV-MNIST. Results are reported in Table 2.

**Results** Still, in the case of mean fusion, confidence-based weighted mean fusion usually doesn't work except in few cases, while our proposed Jacobian regularization exhibits good compatibility with different trained models under various perturbations. The bold results imply that the multi-modal network with our Jacobian regularization invoked outperforms all other models except in FGSM attack ($\omega_4 = 0.03$). Due to page limits, extra results are presented in Appendix B.3.

Table 2: Accuracies of different models (%) are evaluated on RAVDESS when audio features are perturbed. Mean accuracies are reported after repeatedly running 20 times.

| UM / MM | Model | Clean | $\omega_0 = 1.0$ | $\omega_1 = 6$ | $\omega_4 = 0.03$ | $\omega_5 = 0.001$ |
|---|---|---|---|---|---|---|
| UM Nets | 0: Img-regT | 82.5 | 82.5 | 82.5 | 82.5 | 82.5 |
| | 1: Aud-regT | 71.9 | 54.2 | 59.3 | 31.5 | 29.9 |
| | 2: Aud-advT | 78.6 | 43.1 | 71.9 | 53.3 | 62.3 |
| | 3: Aud-freT | 66.4 | 19.5 | 62.4 | 56.0 | 60.5 |
| | 4: Aud-staT | 71.8 | 58.3 | 59.8 | 25.9 | 29.6 |
| | 5: Aud-mixT | 74.6 | 54.5 | 66.9 | 15.7 | 19.2 |
| MM (0, 1) | Mean-w/o | 89.8 | 82.8 | 86.7 | 57.0 | 63.6 |
| | Mean-w/ | 88.5 | 80.4 | 84.1 | 60.7 | 57.3 |
| | Stat-w/o | 89.9 | 82.9 | 86.3 | 56.9 | 63.3 |
| | Stat-w/ (ours) | 89.9 ($\uparrow$) | 85.0 ($\uparrow$) | 87.7 ($\uparrow$) | 59.4 ($\downarrow$) | 68.1 ($\uparrow$) |
| MM (0, 2) | Mean-w/o | 90.8 | 76.9 | 89.5 | 87.3 | 90.6 |
| | Mean-w/ | 89.3 | 76.9 | 87.6 | 79.7 | 85.9 |
| | Stat-w/o | 90.8 | 77.1 | 89.6 | 87.2 | 90.5 |
| | Stat-w/ (ours) | **91.4** ($\uparrow$) | 79.9 ($\uparrow$) | **90.2** ($\uparrow$) | 88.6 ($\uparrow$) | 90.6 ($\uparrow$) |
| MM (0, 3) | Mean-w/o | 90.5 | 60.8 | 89.6 | **90.1** | 90.8 |
| | Mean-w/ | 86.9 | 64.0 | 86.6 | 82.2 | 85.7 |
| | Stat-w/o | 90.5 | 61.5 | 89.2 | 90.0 | 90.7 |
| | Stat-w/ (ours) | 90.8 ($\uparrow$) | 65.6 ($\uparrow$) | 90.0 ($\uparrow$) | 89.5 ($\downarrow$) | **90.7** ($\downarrow$) |
| MM (0, 4) | Mean-w/o | 90.7 | 88.4 | 88.5 | 69.3 | 78.8 |
| | Mean-w/ | 89.2 | 85.4 | 85.5 | 68.9 | 69.3 |
| | Stat-w/o | 90.4 | 88.3 | 88.5 | 69.7 | 78.9 |
| | Stat-w/ (ours) | 90.8 ($\uparrow$) | **88.8** ($\uparrow$) | 88.7 ($\uparrow$) | 75.3 ($\uparrow$) | 82.3 ($\uparrow$) |
| MM (0, 5) | Mean-w/o | 89.5 | 87.8 | 87.9 | 81.7 | 82.7 |
| | Mean-w/ | 87.9 | 85.8 | 86.5 | 82.0 | 79.7 |
| | Stat-w/o | 89.4 | 87.7 | 87.9 | 81.6 | 82.5 |
| | Stat-w/ (ours) | 86.6 ($\downarrow$) | 86.7 ($\downarrow$) | 86.0 ($\downarrow$) | 82.9 ($\uparrow$) | 83.7 ($\uparrow$) |

## 5.3 VGGSOUND

We consider classification task on a real-world data set VGGSound (Chen et al., 2020a), where two modalities audio and video are available. To construct an affordable problem, our classification task only focuses on a subset of all classes. Namely, we randomly choose 100 classes and our classification problem is constrained on them. Consequently, there are 53622 audio-video pairs for training and 4706 audio-video pairs for testing. For the audio modality, we apply a short-time Fourier transform on each raw waveform, generating a $313 \times 513$ spectrogram. We take a 2D ResNet-18 (He et al., 2016) to process these spectrograms. For the video modality, we evenly sample 32 frames from each 10-second video resulting input feature size of $32 \times 256 \times 256$. We take a ResNet-18 network to process the video input where 3D convolutions have been adopted to replace 2D convolutions.

**Results** We present the results of mean fusion and statistical fusion with or without robust add-on in Table 3. Unlike the previous experiments, here we make audio modality clean and assume corruptions/attacks on the video modality. The results demonstrate that our method outperforms counterpart statistical fusion without Jacobian regularization in most cases. Furthermore, the bold results imply that our Jacobian regularization could enhance model robustness under various corruptions or attacks. More results (e.g., unknown corrupted modlity) are presented in Appendix B.4

Table 3: Accuracies of different models (%) are evaluated on VGGSound when video features are perturbed. Mean accuracies are reported after repeatedly running 5 times.

| UM / MM | Model | Clean | $\omega_0 = 1.5$ | $\omega_1 = 6$ | $\omega_4 = 0.03$ | $\omega_5 = 0.001$ |
|---------|-------|-------|------------------|----------------|-------------------|--------------------|
| UM Nets | 0: Aud-regT | 54.4 | 15.0 | 49.8 | 23.0 | 19.9 |
| | 1: Img-regT | 27.4 | 5.8 | 27.4 | 9.5 | 9.0 |
| | 2: Img-advT | 27.5 | 5.3 | 27.4 | 10.7 | 10.3 |
| | 3: Img-freT | 25.2 | 4.0 | 24.2 | 20.4 | 22.9 |
| | 4: Img-staT | 27.0 | 6.9 | 26.9 | 10.5 | 9.6 |
| | 5: Img-mixT | 27.2 | 8.4 | 27.1 | 7.3 | 7.2 |
| MM (0, 1) | Mean-w/o | 57.7 | 45.8 | 57.7 | 35.0 | 25.7 |
| | Mean-w/ | 53.9 | 48.6 | 53.9 | 37.9 | 20.5 |
| | Stat-w/o | 58.5 | 46.0 | 58.4 | 35.3 | 26.0 |
| | Stat-w/ (ours) | 60.1 (↑) | 51.2 (↑) | 60.0 (↑) | 39.7 (↑) | 28.5 (↑) |
| MM (0, 2) | Mean-w/o | 58.8 | 45.0 | 58.8 | 37.2 | 26.9 |
| | Mean-w/ | 54.1 | 49.0 | 54.1 | 38.3 | 21.3 |
| | Stat-w/o | 59.4 | 45.4 | 59.4 | 37.3 | 27.2 |
| | Stat-w/ (ours) | **61.1** (↑) | 50.1 (↑) | **61.1** (↑) | 41.2 (↑) | 29.7 (↑) |
| MM (0, 3) | Mean-w/o | 57.9 | 51.5 | 57.9 | 57.8 | 57.7 |
| | Mean-w/ | 55.2 | 53.6 | 55.2 | 55.0 | 55.2 |
| | Stat-w/o | 58.8 | 52.6 | 58.8 | 55.6 | 57.7 |
| | Stat-w/ (ours) | 56.7 (↓) | 53.2 (↓) | 56.7 (↓) | **58.5** (↑) | **57.9** (↑) |
| MM (0, 4) | Mean-w/o | 57.5 | 47.2 | 57.3 | 36.9 | 30.9 |
| | Mean-w/ | 53.5 | 49.2 | 53.4 | 37.2 | 23.4 |
| | Stat-w/o | 58.2 | 47.5 | 58.1 | 37.4 | 31.0 |
| | Stat-w/ (ours) | 59.8 (↑) | 51.9 (↑) | 59.7 (↑) | 41.0 (↑) | 34.0 (↑) |
| MM (0, 5) | Mean-w/o | 57.8 | 55.3 | 57.8 | 53.9 | 53.5 |
| | Mean-w/ | 55.6 | 54.7 | 55.6 | 54.3 | 49.5 |
| | Stat-w/o | 59.1 | **56.4** | 59.0 | 54.5 | 54.3 |
| | Stat-w/ (ours) | 57.1 (↓) | 55.9 (↓) | 57.0 (↓) | 55.2 (↑) | 55.4 (↑) |

## 6 DISCUSSION AND FUTURE WORK

In this paper, we propose a training-free robust late-fusion method. Intuitively, our method designs a filter for each sample during inference (i.e., test time). Such a filter is implemented by Jacobian regularization, and after a sample goes through it, the change of final multimodal prediction is limited to some amount under an input perturbation. The error bound analysis and series of experiments justify the efficacy of our method both theoretically and numerically. The Twomoon example explains the biasing effect of the extra modality, rooted for the difference between unimodal and multimodal.

Our method opens up other directions for further exploration. In the unimodal context, we understand that directly applying our method usually doesn't change the prediction. Thus, would it be capable to adjust the confidence of a DNN? Another possibility is that besides $\mathbf{W}_x$, we deliberately add another bias term $\mathbf{b}_x$ and optimize both for unimodal robustness. Alternatively, in the multimodal context, we might consider minimizing the derivative of the final prediction directly to the input feature. Moreover, it is educational to compare it with consistency regularization. Last but not least, statistical fusion implicitly uses the assumption that train and test data come from the same distribution, so that we could calculate **freq** based on the train dataset and reuse it in inference time. When domain shift presents, this doesn't work and a straightforward remedy might be to make **freq** a learnable parameter.

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

# A  MISSING DEDUCTION

## A.1  EQUIVALENT FORM AND DERIVATIVE OF STATISTICAL FUSION

In this subsection, we first prove Lemma 1. For conciseness, we denote the $k$-th element of $\mathbf{p}_A$ as $p_{A,k}$. Similar notations are adopted for $\mathbf{p}_B$, $\mathbf{z}_A$, $\mathbf{z}_B$, and $\ln \mathbf{freq}$. The $k$-th element of the left-hand side (LHS) in Lemma 1 can be simplified as:

$$
\begin{aligned}
\frac{p_{A,k}p_{B,k}/freq_k}{\sum_{j=1}^{K} p_{A,j}p_{B,j}/freq_j}
&= \frac{\frac{\exp(z_{A,k})}{T_A}\frac{\exp(z_{B,k})}{T_B}/freq_k}{\sum_{j=1}^{K}\frac{\exp(z_{A,j})}{T_A}\frac{\exp(z_{B,k})}{T_B}/freq_j} \\
&= \frac{\exp(z_{A,k})\exp(z_{B,k})/freq_k}{\sum_{j=1}^{K}\exp(z_{A,j})\exp(z_{B,k})/freq_j} \\
&= \frac{\exp(z_{A,k}+z_{B,k}-\ln freq_k)}{\sum_{j=1}^{K}\exp(z_{A,j}+z_{B,j}-\ln freq_j)}
\end{aligned}
\tag{6}
$$

which is exactly the $k$-th element of the right-hand side (RHS) in Lemma 1 by definition. Note that in the first line of (6), we use the definition $\mathbf{p}_A = \boldsymbol{\sigma}_1(\mathbf{z}_A)$, $\mathbf{p}_B = \boldsymbol{\sigma}_1(\mathbf{z}_B)$, and have introduced two normalization constants $T_A = ||\exp(\mathbf{z}_A)||_1$ and $T_B = ||\exp(\mathbf{z}_B)||_1$.

By analogy to the above deduction, an equivalent form of (3) is $\mathbf{p}' = \boldsymbol{\sigma}_1(\mathbf{W}_a\mathbf{z}_A+\mathbf{W}_b\mathbf{z}_B-\ln \mathbf{freq})$. Based on the chain rule and $\mathbf{z}_A = \mathbf{W}_A\mathbf{h}_A + \mathbf{b}_A$, we have:

$$
\frac{\partial \mathbf{p}'}{\partial \mathbf{h}_A} = \frac{\partial \mathbf{p}'}{\partial \mathbf{z}_A}\frac{\partial \mathbf{z}_A}{\partial \mathbf{h}_A} = \mathbf{J}'\mathbf{W}_a\mathbf{W}_A = [\mathbf{p}'\mathbf{p}'^{,T} - \mathrm{Diag}(\mathbf{p}')]\mathbf{W}_a\mathbf{W}_A
\tag{7}
$$

where we have used the Jacobian matrix of Softmax function to calculate $\mathbf{J}'$ (Bishop, 2006). We emphasize that $\mathbf{J}'$ is symmetric.

## A.2  SOLUTION OF REGULARIZED JACOBIAN LOSS

Because Frobenius norm is convex and a weighted summation with positive coefficients preserves convexity, the relaxed (5) shown in Step 5 of Algorithm 1 is a convex function with respect to $\mathbf{W}_a$. This implies that any local minimum is also the global minimum. We calculate the derivative of the relaxed (5) with respect to $\mathbf{W}_a$ and set it to zero, yielding:

$$
\begin{aligned}
(\frac{1}{\gamma} - 1)\mathbf{J}^{(t),2}\mathbf{W}_a(\mathbf{W}_A\mathbf{W}_A^T) + \mathbf{W}_a &= \mathbf{I} \\
\Rightarrow \mathbf{A}\mathbf{W}_a + \mathbf{W}_a\mathbf{B} &= \mathbf{B}
\end{aligned}
\tag{8}
$$

where for simplicity we have defined $\mathbf{A} = (1/\gamma - 1)\mathbf{J}^{(t),2} \in \mathbb{R}^{K \times K}$ and $\mathbf{B} = (\mathbf{W}_A\mathbf{W}_A^T)^{-1} \in \mathbb{R}^{K \times K}$. (8) is known as Sylvester equation[2], and it has been proven that if $\mathbf{A}$ and $-\mathbf{B}$ do not share any eigenvalues, then the Sylvester equation has an unique solution (Jameson, 1968; Horn & Johnson, 2012; Bartels & Stewart, 1972). In our case, since $\mathbf{A}$ is positive semi-definite (PSD) and $-\mathbf{B}$ is negative semi-definite, the only possible overlapping eigenvalue is $0$. On one hand, $0$ is surely an eigenvalue of $\mathbf{A}$ since $\mathbf{J}\mathbf{u} = 0\mathbf{u}$, where $\mathbf{u}$ is a column vector with all elements equal to $1$. On the other hand, it is argued that a well-generalized neural network normally doesn't have singular weight matrices (Martin & Mahoney, 2021). Thus, $0$ is almost surely not the overlapping eigenvalue and a unique solution of (8) is guaranteed. As demonstrated by Jameson (1968), the matrix equation (8) can be converted to invert a $K \times K$ matrix. As in a classification task, $K$ won't be too large, the run time of solving Sylvester equation is affordable in this context. Furthermore, as shown in the above equation, when $\gamma \to 0$, the matrix $A$ will tend to infinity, leading to an ill-conditioned matrix equation. This, in turn, implies that the second term in the loss shown in Eq (5) guarantees numerical stability.

---

[2]A general Sylvester equation has the form: $\mathbf{A}\mathbf{X} + \mathbf{X}\mathbf{B} = \mathbf{C}$, where $\{\mathbf{A}, \mathbf{B}, \mathbf{C}\}$ are all known and $\mathbf{X}$ is required to be solved.

### A.3 Proof of Theorem 1

In the last iteration $t = t_{\max} \to \infty$, noticing $\mathbf{W}_a$ satisfies (8), we have:

$$
\begin{aligned}
||\mathbf{J}^{(t_{\max})}\mathbf{W}_a\mathbf{W}_A||_F^2 &= \text{Tr}[\mathbf{J}^{(t_{\max})}\mathbf{W}_a\mathbf{W}_A\mathbf{W}_A^T\mathbf{W}_a^T\mathbf{J}^{(t_{\max})}] \\
&= \text{Tr}[\mathbf{J}^{(t_{\max}),2}\mathbf{W}_a(\mathbf{W}_A\mathbf{W}_A^T)\mathbf{W}_a^T] \\
&= (\frac{1}{\gamma}-1)^{-1}\text{Tr}[\mathbf{A}\mathbf{W}_a\mathbf{B}^{-1}\mathbf{W}_a^T] \\
&= (\frac{1}{\gamma}-1)^{-1}\text{Tr}[(\mathbf{B}-\mathbf{W}_a\mathbf{B})\mathbf{B}^{-1}\mathbf{W}_a^T] \\
&= (\frac{1}{\gamma}-1)^{-1}\text{Tr}[\mathbf{W}_a-\mathbf{W}_a\mathbf{W}_a^T]
\end{aligned}
\tag{9}
$$

and:

$$
||\mathbf{W}_a - \mathbf{I}||_F^2 = \text{Tr}[\mathbf{W}_a\mathbf{W}_a^T - 2\mathbf{W}_a + \mathbf{I}]
\tag{10}
$$

Thus we have:

$$
\begin{aligned}
||\mathbf{J}^{(t_{\max})}\mathbf{W}_a\mathbf{W}_A||_F^2 &\leq \frac{1}{1-\gamma}[(1-\gamma)||\mathbf{J}^{(t_{\max})}\mathbf{W}_a\mathbf{W}_A||_F^2 + \frac{\gamma}{2}||\mathbf{W}_a-\mathbf{I}||_F^2] \\
&= \frac{1}{1-\gamma}\left\{\gamma\text{Tr}[\mathbf{W}_a-\mathbf{W}_a\mathbf{W}_a^T] + \frac{\gamma}{2}\text{Tr}[\mathbf{W}_a\mathbf{W}_a^T - 2\mathbf{W}_a^T + \mathbf{I}]\right\} \\
&= \frac{\gamma}{2(1-\gamma)}\text{Tr}[\mathbf{I}-\mathbf{W}_a\mathbf{W}_a^T] \\
&\leq \frac{\gamma}{2(1-\gamma)}\text{Tr}[\mathbf{I}] = \frac{\gamma K}{2(1-\gamma)}
\end{aligned}
\tag{11}
$$

where we have taken advantage of the fact that $\mathbf{W}_a\mathbf{W}_a^T$ is a positive semi-definite matrix, and thus its trace is no less than zero. Based on the first-order Taylor expansion, we have:

$$
\begin{aligned}
\mathbf{p}^{\prime,noise} - \mathbf{p}' &\approx \frac{\partial \mathbf{p}'}{\partial \mathbf{h}_A}(\mathbf{h}_A^{\text{noise}} - \mathbf{h}_A) \\
&= \frac{\partial \mathbf{p}^{(t_{\max}+1)}}{\partial \mathbf{h}_A}(\mathbf{h}_A^{\text{noise}} - \mathbf{h}_A) \\
&\approx \mathbf{J}^{(t_{\max})}\mathbf{W}_a\mathbf{W}_A(\mathbf{h}_A^{\text{noise}} - \mathbf{h}_A)
\end{aligned}
\tag{12}
$$

Thus we have:

$$
\begin{aligned}
||\mathbf{p}^{\prime,noise} - \mathbf{p}'|| &\leq ||\mathbf{J}^{(t_{\max})}\mathbf{W}_a\mathbf{W}_A||_F \cdot ||\mathbf{h}_A^{\text{noise}} - \mathbf{h}_A|| \\
&\leq \sqrt{\frac{\gamma K}{2(1-\gamma)}} \cdot l||\mathbf{x}_A^{\text{noise}} - \mathbf{x}_A|| \\
&= l\sqrt{\frac{\gamma K}{2(1-\gamma)}}||\boldsymbol{\epsilon}||
\end{aligned}
\tag{13}
$$

where we have utilized the $l$-Lipschitz continuity of $\mathbf{f}_A$.

### A.4 Unimodal 2D Case

Consider an illustrating example when $K = 2$ (a.k.a. unimodal binary classification). In this case, the final prediction is two dimensional: $\mathbf{p} = [p_1\, p_2]^T$. We denote the raw logit after adding the $\mathbf{W}_x$ matrix as $\mathbf{z}' = \mathbf{W}_x\mathbf{z}$, and its entries are denoted as $z_1'$ and $z_2'$. Let us begin by explicitly writing out the first line of (8). After some simplification, we obtain:

$$
\begin{bmatrix} \kappa & -\kappa \\ -\kappa & \kappa \end{bmatrix}\begin{bmatrix} \rho_1 & \rho_2 \\ \rho_3 & \rho_4 \end{bmatrix}\begin{bmatrix} a & b \\ b & c \end{bmatrix} + \begin{bmatrix} \rho_1 & \rho_2 \\ \rho_3 & \rho_4 \end{bmatrix} = \mathbf{I}
\tag{14}
$$

where $\kappa = 1/\gamma - 1$. Here we have explicitly written out the elements of $\mathbf{W}\mathbf{W}^T$ and merged some terms of $\mathbf{J}'^{,2}$ into $a = p_1p_2(w_{11}^2 + w_{12}^2)$, $b = p_1p_2(w_{21}w_{11} + w_{12}w_{22})$, and $c = p_1p_2(w_{21}^2 + w_{22}^2)$.

The unknowns occur in $\mathbf{W}_x$:

$$\mathbf{W}_x = \begin{bmatrix} \rho_1 & \rho_2 \\ \rho_3 & \rho_4 \end{bmatrix} \tag{15}$$

After some algebra, we can solve all unknowns via the following block matrix inversion:

$$\begin{bmatrix} \rho_1 \\ \rho_2 \\ \rho_3 \\ \rho_4 \end{bmatrix} = \begin{bmatrix} \mathbf{G} & \mathbf{H} \\ \mathbf{H} & \mathbf{G} \end{bmatrix}^{-1} \cdot \begin{bmatrix} 1 \\ 0 \\ 0 \\ 1 \end{bmatrix} \tag{16}$$

where for simplicity, we have defined:

$$\mathbf{G} = \begin{bmatrix} 1 + a\kappa & b\kappa \\ b\kappa & 1 + c\kappa \end{bmatrix}, \quad \mathbf{H} = \begin{bmatrix} -a\kappa & -b\kappa \\ -b\kappa & -c\kappa \end{bmatrix} \tag{17}$$

Noticing that $\mathbf{z}' = \mathbf{W}_x \mathbf{z}$, we can prove that:

$$(z_1' - z_2') \cdot (z_1 - z_2) = \mathbf{z}^T \begin{bmatrix} \rho_1 - \rho_3 & -(\rho_1 - \rho_3) \\ \rho_2 - \rho_4 & -(\rho_2 - \rho_4) \end{bmatrix} \mathbf{z} \tag{18}$$

It is easy to calculate the eigenvalue of the matrix in the middle is 0 and $\rho_1 + \rho_4 - \rho_2 - \rho_3$. In other words, if we could prove $\rho_1 + \rho_4 - \rho_2 - \rho_3$ is no less than zero, then the matrix in the middle is positive semi-definite, which implies order preserving property holds: If $z_1 > z_2$, then $z_1' > z_2'$ and vice versa. This order preserving property is equivalent to say that adding the $\mathbf{W}_x$ matrix makes no sense in unimodal binary classification problem, since the predication won't change for any input.

Now let us prove $\rho_1 + \rho_4 - \rho_2 - \rho_3 \geq 0$. With a lot of algebra, we obtain:

$$\begin{aligned} \rho_1 + \rho_4 - \rho_2 - \rho_3 &= \begin{bmatrix} 1 & -1 & -1 & 1 \end{bmatrix} \begin{bmatrix} \rho_1 \\ \rho_2 \\ \rho_3 \\ \rho_4 \end{bmatrix} \\ &= \begin{bmatrix} 1 & -1 & -1 & 1 \end{bmatrix} \begin{bmatrix} \mathbf{G} & \mathbf{H} \\ \mathbf{H} & \mathbf{G} \end{bmatrix}^{-1} \cdot \begin{bmatrix} 1 \\ 0 \\ 0 \\ 1 \end{bmatrix} \\ &= \frac{2(a + 2b + c)\kappa + 2}{4(ac - b^2)\kappa^2 + 2(a + c)\kappa + 1} \end{aligned} \tag{19}$$

Based on the definitions of $\{a, b, c, d\}$, by using Cauchy inequality and completing square, we could prove: $ac - b^2 \geq 0$, $a + c \geq 0$, $a + 2b + c \geq 0$, and $a - 2b + c \geq 0$.

Treat $\kappa \in (0, \infty)$ as a variable. The derivative of (19) with respect to $\kappa$ is a fraction. Its denominator is always larger than zero, while the numerator is a quadratic polynomial:

$$\underbrace{8(b^2 - ac)(a + 2b + c)}_{\leq 0} \kappa^2 + \underbrace{16(b^2 - ac)}_{\leq 0} \kappa - 2 \underbrace{(a - 2b + c)}_{\geq 0} \tag{20}$$

Thus $\rho_1 + \rho_4 - \rho_2 - \rho_3$ monotonically decreases when $\kappa$ increases in $(0, \infty)$. Its minimum value equals 0 and is attained when $\kappa \to \infty$. Our proof completes.

### A.5 Unimodal High-Dimensional Case

Without loss of generality, we consider the case when $\mathbf{W}\mathbf{W}^T = \mathbf{I}$. We emphasize this is a mild assumption. Because when $\mathbf{W}$ doesn't satisfy this condition, we could perform SVD decomposition $\mathbf{W} = \mathbf{U}\boldsymbol{\Sigma}\mathbf{V}$ and convert the FC layer $\mathbf{W}\mathbf{h} + \mathbf{b}$ into three sequential linear layers: $\mathbf{h} \to \mathbf{V}\mathbf{h} \to \boldsymbol{\Sigma}\mathbf{V}\mathbf{h} \to \mathbf{U}\boldsymbol{\Sigma}\mathbf{V}\mathbf{h} + \mathbf{b}$, where now the weight matrix in the last layer satisfies $\mathbf{U}\mathbf{U}^T = \mathbf{I}$. Under this assumption $\mathbf{B} = (\mathbf{W}\mathbf{W}^T)^{-1} = \mathbf{I}$, (8) could be further simplified as:

$$\mathbf{W}_x = [(\frac{1}{\gamma} - 1)\mathbf{J}'^{,2} + \mathbf{I}]^{-1} = (\kappa \mathbf{J}'^{,2} + \mathbf{I})^{-1} \tag{21}$$

where $\mathbf{J}' = \mathbf{J}^{(0)}$ is known, and for simplicity, we have denoted $\kappa = 1/\gamma - 1$. From this expression, it is clear that the second term of (5) guarantees numerical stability, since $\mathbf{J}'^{,2}$ is not invertible and without the identity matrix (21) doesn't exist. Before moving on, we define a helper vector $\mathbf{e}_{ij} \in \mathbb{R}^K$, whose $i$-th and $j$-th entries equal 1 and $-1$, respectively, and all other entries equal 0. Then considering the relation of $\mathbf{z}' = \mathbf{W}_x \mathbf{z}$, we have:

$$
\begin{aligned}
(z'_i - z'_j)(z_i - z_j) &= (\mathbf{z}'^{,T} \mathbf{e}_{ij})(\mathbf{e}_{ij}^T \mathbf{z}) \\
&= \mathbf{z}'^{,T} \mathbf{e}_{ij} \mathbf{e}_{ij}^T \mathbf{W}_x^{-1} \mathbf{z}'
\end{aligned}
\tag{22}
$$

This implies that if $\mathbf{e}_{ij}\mathbf{e}_{ij}^T \mathbf{W}_x^{-1}$ is a positive semi-definite (PSD) matrix for arbitrary $i \neq j$, then the order preserving property holds (i.e., if $z_i > z_j$, then $z'_i > z'_j$). Since $\mathbf{e}_{ij}\mathbf{e}_{ij}^T \mathbf{W}_x^{-1}$ is asymmetrical, examining whether it is PSD requires us to focus on the summation of its transpose and itself. Namely, $\mathbf{e}_{ij}\mathbf{e}_{ij}^T \mathbf{W}_x^{-1}$ is PSD if and only if the eigenvalues of $[\mathbf{e}_{ij}\mathbf{e}_{ij}^T \mathbf{W}_x^{-1} + (\mathbf{e}_{ij}\mathbf{e}_{ij}^T \mathbf{W}_x^{-1})^T]$ are all no less than zero. Notice that $\mathbf{W}_x$ and $\mathbf{W}_x^{-1}$ are symmetric as shown in (21), we have

$$
\begin{aligned}
\mathbf{e}_{ij}\mathbf{e}_{ij}^T \mathbf{W}_x^{-1} + (\mathbf{e}_{ij}\mathbf{e}_{ij}^T \mathbf{W}_x^{-1})^T &= \mathbf{e}_{ij}\mathbf{e}_{ij}^T \mathbf{W}_x^{-1} + \mathbf{W}_x^{-1}\mathbf{e}_{ij}\mathbf{e}_{ij}^T \\
&= \mathbf{e}_{ij}\mathbf{e}_{ij}^T(\kappa \mathbf{J}'^{,2} + \mathbf{I}) + (\kappa \mathbf{J}'^{,2} + \mathbf{I})\mathbf{e}_{ij}\mathbf{e}_{ij}^T \\
&= 2\mathbf{e}_{ij}\mathbf{e}_{ij}^T + \kappa(\mathbf{e}_{ij}\mathbf{e}_{ij}^T \mathbf{J}'^{,2} + \mathbf{J}'^{,2}\mathbf{e}_{ij}\mathbf{e}_{ij}^T)
\end{aligned}
\tag{23}
$$

For simplicity we denote the set of $\kappa$ which can make all eigenvalues of the aforementioned matrix non-negative as: $\Gamma_{ij} = \{\kappa > 0 | \lambda_{min}(2\mathbf{e}_{ij}\mathbf{e}_{ij}^T + \kappa(\mathbf{e}_{ij}\mathbf{e}_{ij}^T \mathbf{J}'^{,2} + \mathbf{J}'^{,2}\mathbf{e}_{ij}\mathbf{e}_{ij}^T)) \geq 0\}$, where $\lambda_{min}(\cdot)$ represents the minimum eigenvalue of a matrix. Then a necessary and sufficient condition for the order persevering property is $\kappa \in \cap_{i \neq j} \Gamma_{ij}$. An immediate corollary is: if we want the final prediction to change after adding $\mathbf{W}_x$, then we must have $\frac{1}{\gamma} - 1 \notin \cap_{i \neq j} \Gamma_{ij}$.

# B EXPERIMENT DETAILS AND ABLATION STUDY

## B.1 DEFINITION OF OMEGA

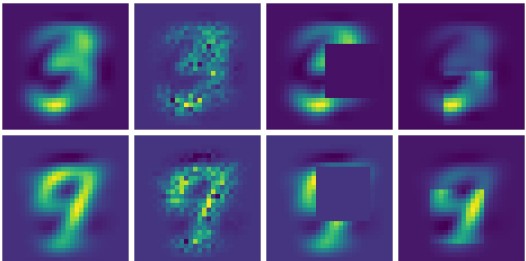

Figure 5: Image samples in AV-MNIST. Original images are shown in the first column, and their counterparts perturbed by Gaussian noise ($\omega_0 = 0.2$) are shown in the second column. The third ($\omega_3 = -1$) and last ($\omega_3 = 2$) column show the images perturbed by bias noise.

Take AV-MNIST as an example. We have defined the following types of perturbations.

- **Gaussian noise**: Perturb every element $x$ in the image (or audio) feature by a Gaussian noise in the form of $x^{noise} = (1 + \epsilon)x$, where $\epsilon \sim N(0, \omega_0^2)$.

- **Missing entries**: For the audio modality, we randomly select $\omega_1$ consecutive columns (or rows) and all elements in there are set to 0. This corresponds to missing time frames (or frequency components).

- **Bias noise**: For the image modality, we randomly select an image patch with size of $\omega_2 \times \omega_2$, and every element $x$ in there is perturbed by a given amount: $x^{noise} = (1 + \omega_3)x$. This corresponds to change of illumination. (see Figure 5).

- **FGSM attack**: We use the Cleverhans libarary (Papernot et al., 2016) to implement a FGSM attack on the input image/audio feature with $l_\infty$ norm. The maximum overall norm variation is $\omega_4$ (e.g., $\omega_4 = 0.3$).

- **PGD attack**: Cleverhans is adopted to implement a PGD attack on the input image/audio feature with $l_\infty$ norm. The maximum overall norm variation also equals $\omega_4$, and that for each inner attack is $\omega_5$ (e.g., $\omega_5 = 0.01$), and the number of inner attacks is $\omega_6$.

Unless explicitly mentioned, $\omega_6$ is set to 20.

## B.2 EXTRA RESULTS ON AV-MNIST

**Impact of hyper-parameters** Following Figure 4, we plot model accuracy versus magnitude of noise/attack with different gamma values in Figure 6. For comparison purpose, Figure 4 is re-drawn in its first row. As demonstrated in Figure 6, alongside the increasing of $\gamma$, the gap between the orange and blue lines becomes smaller. However, the orange lines consistently appear upon the blue lines, indicating that our method works in a wide range of gamma value and that hyper-parameter tuning is relatively easy in our method. We emphasize that $\gamma = 1.0$ is equivalent to trivial fusion.

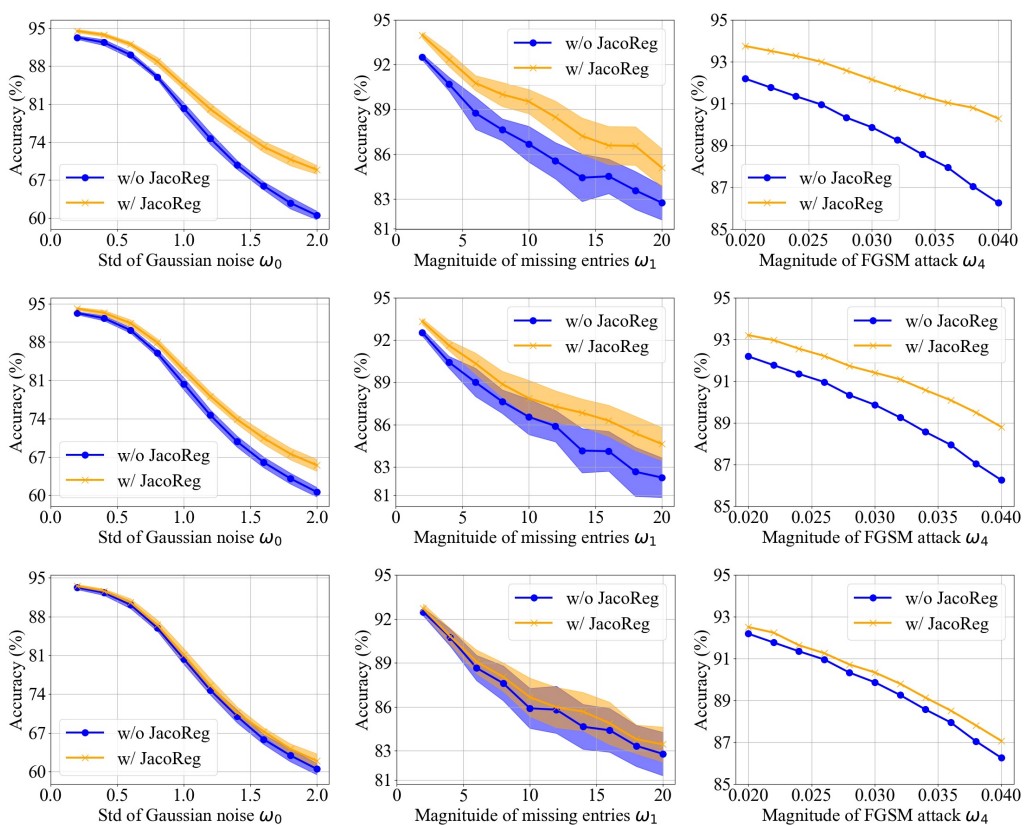

Figure 6: Accuracies of multimodal networks obtained by statistically fusing a vanilla image network and audio network with $\gamma = 0.1$ (the 1-st row), $\gamma = 0.5$ (the 2-nd row), and $\gamma = 0.9$ (the 3-rd row). Mean and Std of 20 repeated experiments are shown by the solid lines and the shaded regions, respectively. Note that FGSM attack is deterministic, thus there is almost no variance when repeating experiments.

**Perturbation on the image modality** Unlike in the main text, here we consider perturbations on the image modality, while the audio modality is assumed clean. Consequently, we invoke our Jacobian regularization method for the image modality, while keep the audio modality unchanged. We have deliberately enlarged the magnitude of the noise/attack, and results are reported in Table 4.

As shown in Table 4, confidence-weighted summation still doesn't outperform purely mean fusion in all cases. Regards to statistical fusion, the second column demonstrates that this time our Jacobian regularization might lead to accuracy drop on clean data. The phenomenon that robust network

is less accurate on clean data might happen as suggested by Tsipras et al. (2019). However, we notice that such phenomenon doesn't occur in Table 1. We believe that such difference might lie intrinsically in modality difference, and we will explore it in our future work.

Table 4: Accuracies of different models (%) are evaluated on AV-MNIST when image features are perturbed. Mean accuracies are reported after repeatedly running 20 times.

| UM / MM | Model | Clean | $\omega_0 = 2.5$ | $\omega_{2,3} = 10, 2$ | $\omega_4 = 0.07$ | $\omega_5 = 0.008$ |
|---|---|---|---|---|---|---|
| UM Nets | 0: Aud-regT | 83.9 | 83.9 | 83.9 | 83.9 | 83.9 |
| | 1: Img-regT | 73.4 | 24.4 | 37.2 | 15.6 | 12.3 |
| | 2: Img-advT | 73.1 | 33.4 | 40.5 | 43.3 | 34.1 |
| | 3: Img-freT | 65.1 | 36.2 | 40.7 | 46.7 | 42.6 |
| | 4: Img-staT | 74.2 | 29.5 | 49.0 | 19.4 | 14.1 |
| | 5: Img-mixT | 74.1 | 30.4 | 37.6 | 37.3 | 23.9 |
| MM (0, 1) | Mean-w/o | 93.6 | 71.9 | 84.6 | 83.2 | 80.0 |
| | Mean-w/ | 90.5 | 78.1 | 83.5 | 81.6 | 77.7 |
| | Stat-w/o | 93.6 | 71.7 | 84.2 | 83.1 | 79.9 |
| | Stat-w/ | 91.3 ($\downarrow$) | 75.2 ($\downarrow$) | 86.4 ($\uparrow$) | 85.4 ($\uparrow$) | 83.3 ($\uparrow$) |
| MM (0, 2) | Mean-w/o | **93.9** | 83.6 | 85.4 | 89.9 | 88.4 |
| | Mean-w/ | 91.5 | 82.7 | 82.0 | 86.1 | 83.2 |
| | Stat-w/o | **93.9** | 83.5 | 85.4 | **89.9** | 88.4 |
| | Stat-w/ (ours) | 90.8 ($\downarrow$) | 85.0 ($\uparrow$) | 86.9 ($\uparrow$) | 88.1 ($\downarrow$) | 87.6 ($\downarrow$) |
| MM (0, 3) | Mean-w/o | 91.4 | 87.1 | 88.1 | 88.8 | **88.6** |
| | Mean-w/ | 88.3 | 85.6 | 85.7 | 85.9 | 85.5 |
| | Stat-w/o | 91.4 | 87.1 | 88.2 | 88.8 | **88.6** |
| | Stat-w/ (ours) | 90.9 ($\downarrow$) | **87.0** ($\downarrow$) | 88.2 ($\uparrow$) | 88.6 ($\downarrow$) | 88.3 ($\downarrow$) |
| MM (0, 4) | Mean-w/o | 93.7 | 77.8 | 89.0 | 85.1 | 82.0 |
| | Mean-w/ | 91.1 | 83.3 | 86.8 | 82.3 | 78.0 |
| | Stat-w/o | 93.7 | 77.8 | 89.1 | 85.0 | 81.8 |
| | Stat-w/ (ours) | 91.3 ($\downarrow$) | 79.9 ($\downarrow$) | **89.3** ($\uparrow$) | 86.4 ($\uparrow$) | 84.6 ($\uparrow$) |
| MM (0, 5) | Mean-w/o | 92.6 | 85.7 | 86.5 | 87.2 | 84.2 |
| | Mean-w/ | 90.1 | 84.5 | 84.4 | 84.3 | 80.0 |
| | Stat-w/o | 92.6 | 85.7 | 86.7 | 87.2 | 84.1 |
| | Stat-w/ (ours) | 92.2 ($\downarrow$) | 85.7 ($\uparrow$) | 86.9 ($\uparrow$) | 87.1 ($\downarrow$) | 84.6 ($\uparrow$) |

**Noise on both modality** We further consider the setting when both image and audio modality are perturbed by Gaussian noise in Table 5, which differs ours from Kim & Ghosh (2019).

### B.3 EXTRA RESULTS ON RAVDESS

**Impact of hyper-parameters** Following Figure 6, we also plot model accuracy versus magnitude of noise/attack with different gamma values in the emotion recognition experiment. The results are shown in Figure 7. The multimodal network with our Jacobian regularization invoked (i.e., the orange line) almost surpasses the network without that invoked (i.e., the blue line) in all cases.

**Perturbation on the image modality** Here we consider perturbations on the image modality. Similarly, we have deliberately enlarged the magnitude of the noise/attack, and results are presented in Table 6. As demonstrated in the table, the mutimodal network enhanced by our method usually achieves the best accuracy among all models except in few cases. Also, our Jacobian regularization could usually improve model accuracy except in the clean case. We notice that in some extreme cases when the image modality is severely perturbed, the multimdoal network might be even worse than the regular unimodal audio network. This, in turn, implies that multimodal fusion is not always the winner, especially when large noise/attack is presented.

### B.4 EXTRA RESULTS ON VGGSOUND

**Impact of hyper-parameters** We also plot model accuracy versus magnitude of noise on video modality with different gamma values in the VGGSound experiment in Figure 8.

**Unknown Noise on Single Modality** To capture another setting, here we consider that the video modality (or audio modality) is corrupted, and both modalities are trained with robust add-ons in Ta-

Table 5: Accuracies of different models (%) are evaluated on AV-MNIST when both modalities are perturbed by Gaussian noise. Mean accuracies are reported after repeatedly running 20 times.

| UM / MM | Model | Clean | $\omega_0 = 1.0$ | $\omega_0 = 1.5$ | $\omega_0 = 2.0$ | $\omega_0 = 2.5$ |
|---|---|---|---|---|---|---|
| UM Nets | 0: Img-regT | 73.4 | 46.7 | 35.9 | 28.8 | 24.2 |
| | 1: Aud-regT | 83.9 | 55.0 | 34.8 | 25.0 | 20.4 |
| | 2: Img-staT | 74.4 | 60.5 | 47.6 | 38.2 | 30.8 |
| | 3: Aud-staT | 85.1 | 70.4 | 51.8 | 37.6 | 28.9 |
| | 4: Img-mixT | 74.1 | 53.0 | 43.3 | 36.0 | 30.4 |
| | 5: Aud-mixT | 87.6 | 61.2 | 41.0 | 28.0 | 21.1 |
| MM (0, 1) | Mean-w/o | 93.6 | 69.6 | 46.8 | 34.0 | 27.5 |
| | Mean-w/ | 91.0 | 64.0 | 43.4 | 32.0 | 25.8 |
| | Stat-w/o | 93.6 | 69.5 | 46.6 | 33.8 | 27.1 |
| | Stat-w/ | 93.5 (↓) | 70.3 (↑) | 47.9 (↑) | 34.6 (↑) | 27.6 (↑) |
| MM (2, 3) | Mean-w/o | 93.0 | 83.1 | 65.4 | 48.5 | 36.6 |
| | Mean-w/ | 90.6 | 78.5 | 60.8 | 46.0 | 36.0 |
| | Stat-w/o | 93.1 | 83.0 | 65.5 | 48.3 | 36.8 |
| | Stat-w/ (ours) | 93.6 (↑) | **84.4** (↑) | **66.7** (↑) | **49.2** (↑) | **36.9** (↑) |
| MM (4, 5) | Mean-w/o | **95.5** | 75.4 | 55.1 | 40.3 | 31.6 |
| | Mean-w/ | 94.1 | 70.5 | 50.1 | 37.6 | 29.8 |
| | Stat-w/o | 95.4 | 75.4 | 54.9 | 40.2 | 31.5 |
| | Stat-w/ (ours) | 95.4 (↓) | 75.2 (↓) | 54.9 (↑) | 40.2 (↓) | 31.6 (↑) |

Table 6: Accuracies of different models (%) are evaluated on RAVDESS when image features are perturbed. Mean accuracies are reported after repeatedly running 20 times.

| UM / MM | Model | Clean | $\omega_0 = 4.0$ | $\omega_{2,3} = 200, -4$ | $\omega_4 = 0.07$ | $\omega_5 = 0.008$ |
|---|---|---|---|---|---|---|
| UM Nets | 0: Aud-regT | 71.9 | 71.9 | 71.9 | 71.9 | 71.9 |
| | 1: Img-regT | 82.5 | 25.9 | 21.3 | 21.1 | 10.7 |
| | 2: Img-advT | 83.3 | 21.1 | 29.4 | 40.0 | 25.1 |
| | 3: Img-freT | 76.0 | 15.3 | 24.7 | 50.1 | 38.7 |
| | 4: Img-staT | 83.7 | 35.1 | 23.7 | 17.0 | 10.9 |
| | 5: Img-mixT | 85.1 | 15.7 | 27.3 | 8.4 | 8.5 |
| MM (0, 1) | Mean-w/o | 89.8 | 63.3 | 40.8 | 49.2 | 21.8 |
| | Mean-w/ | 88.5 | 64.6 | 40.4 | 43.3 | 16.3 |
| | Stat-w/o | 89.9 | 63.6 | 41.1 | 49.4 | 21.8 |
| | Stat-w/ (ours) | 88.7 (↓) | 66.1 (↑) | 43.4 (↑) | 54.5 (↑) | 23.5 (↑) |
| MM (0, 2) | Mean-w/o | 89.2 | 51.8 | 62.0 | 78.2 | 74.0 |
| | Mean-w/ | 87.4 | 61.9 | 63.1 | 69.3 | 56.4 |
| | Stat-w/o | 89.2 | 51.7 | 62.3 | 78.1 | 74.2 |
| | Stat-w/ (ours) | 87.7 (↓) | 54.3 (↑) | 64.6 (↑) | 78.9 (↑) | 75.1 (↑) |
| MM (0, 3) | Mean-w/o | 86.9 | 37.5 | 63.5 | 81.3 | 77.6 |
| | Mean-w/ | 84.1 | 40.1 | 62.0 | 74.6 | 68.1 |
| | Stat-w/o | 86.9 | 37.6 | 63.7 | **81.1** | 77.7 |
| | Stat-w/ (ours) | 84.8 (↓) | 40.9 (↑) | **66.5** (↑) | 80.6 (↓) | **78.6**(↑) |
| MM (0, 4) | Mean-w/o | 89.6 | 75.7 | 52.8 | 55.8 | 32.0 |
| | Mean-w/ | 86.7 | 71.8 | 53.7 | 51.4 | 19.8 |
| | Stat-w/o | **89.5** | 76.3 | 52.9 | 55.7 | 31.7 |
| | Stat-w/ (ours) | 87.8 (↓) | **76.4** (↑) | 56.2 (↑) | 60.5 (↑) | 35.3 (↑) |
| MM (0, 5) | Mean-w/o | 83.0 | 73.1 | 72.0 | 66.5 | 53.8 |
| | Mean-w/ | 82.4 | 69.5 | 69.9 | 66.3 | 33.2 |
| | Stat-w/o | 82.9 | 73.1 | 72.1 | 66.5 | 54.0 |
| | Stat-w/ (ours) | 80.3 (↓) | 73.7 (↑) | 73.3 (↑) | 69.4 (↑) | 58.8 (↑) |

ble 7 and 8, respectively. We emphasize that this setting is valid because in real-world applications, we sometimes do not know which modality is corrupted. As shown in these tables, the multimodal networks fused by our method could usually achieve higher accuracies in most cases, except when

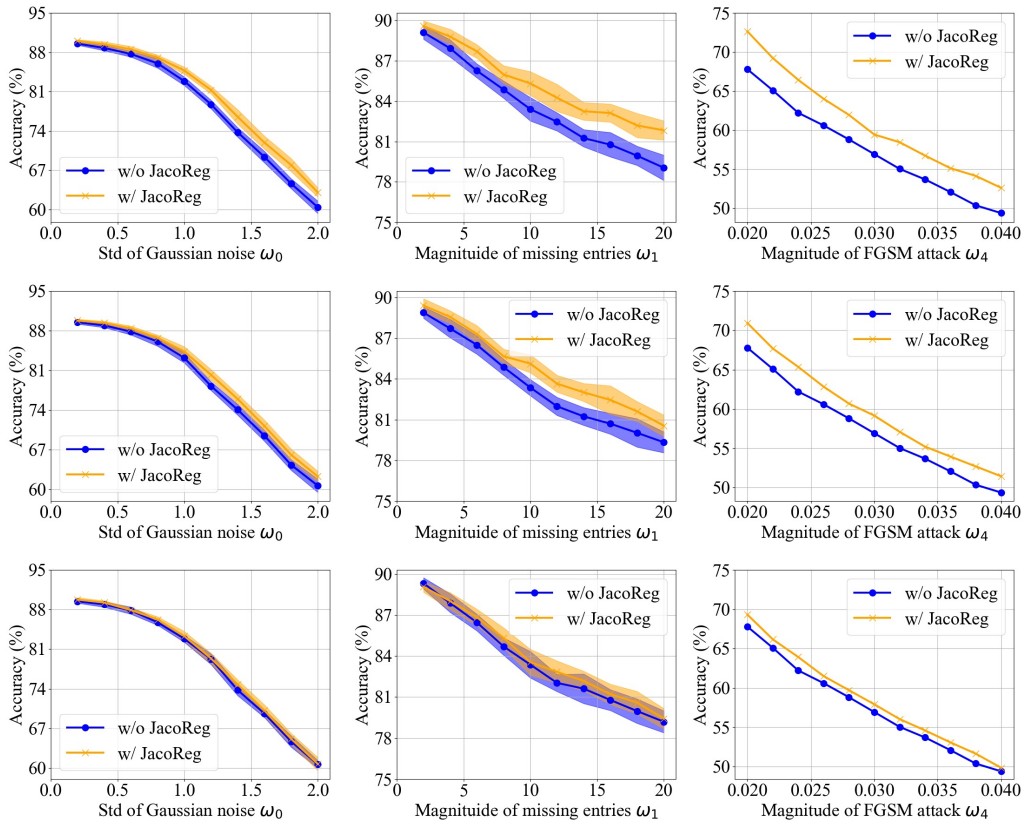

Figure 7: Accuracies of multimodal networks obtained by statistically fusing a vanilla image network and audio network with $\gamma = 0.1$ (the 1-st row), $\gamma = 0.5$ (the 2-nd row), and $\gamma = 0.9$ (the 3-rd row). Mean and Std of 20 repeated experiments are shown by the solid lines and the shaded regions, respectively. Note that FGSM attack is deterministic, thus there is almost no variance when repeating experiments.

both unimodal networks are learned with free-m training [3]. Moreover, the fact that our Jacobian regularized multimodal network with a corrupted modality still outperforms the unimodal network with clean data in all cases demonstrates the positive effect of an extra modality.

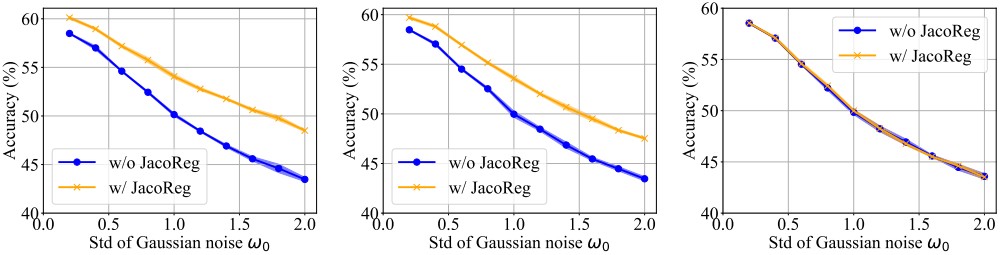

Figure 8: Accuracies of multimodal networks obtained by statistically fusing a vanilla image network and audio network with $\gamma = 0.1$ (the 1-st column), $\gamma = 0.5$ (the 2-nd column), and $\gamma = 0.99$ (the 3-rd column). Mean and Std of 5 repeated experiments are shown by the solid lines and the shaded regions, respectively.

---

[3]We neglect the cases when networks are tested on clean data, since it is possible that a robust network is less accurate on clean data as suggested by Tsipras et al. (2019).

Table 7: Accuracies of different models (%) are evaluated on VGGSound when video features are perturbed. Mean accuracies are reported after repeatedly running 5 times.

| UM / MM | Model | Clean | $\omega_0 = 1.5$ | $\omega_1 = 6$ | $\omega_4 = 0.03$ | $\omega_5 = 0.001$ |
|---|---|---|---|---|---|---|
| | 0: Img-regT | 27.4 | 5.78 | 27.4 | 9.5 | 9.0 |
| | 1: Aud-regT | 54.4 | 15.0 | 49.8 | 23.0 | 19.9 |
| | 2: Img-advT | 27.5 | 5.3 | 27.4 | 10.7 | 10.3 |
| | 3: Aud-advT | 54.8 | 19.4 | 50.6 | 41.8 | 48.5 |
| UM | 4: Img-freT | 22.2 | 4.0 | 22.2 | 19.4 | 20.9 |
| Nets | 5: Aud-freT | 49.2 | 32.4 | 46.6 | 48.2 | 49.3 |
| | 6: Img-staT | 27.0 | 6.9 | 26.9 | 10.5 | 9.6 |
| | 7: Aud-staT | 55.2 | 18.8 | 50.0 | 22.4 | 20.1 |
| | 8: Img-mixT | 27.2 | 8.4 | 27.1 | 7.3 | 7.2 |
| | 9: Aud-mixT | 56.3 | 1.57 | 50.1 | 16.9 | 11.7 |
| | Mean-w/o | 57.7 | 45.8 | 57.6 | 35.0 | 25.7 |
| MM | Mean-w/ | 53.9 | 48.6 | 53.8 | 37.9 | 20.5 |
| (0, 1) | Stat-w/o | 58.5 | 46.0 | 58.4 | 35.3 | 26.0 |
| | Stat-w/ (ours) | 60.1 (↑) | 51.2 (↑) | 60.0 (↑) | 39.7 (↑) | 28.5 (↑) |
| | Mean-w/o | 59.1 | 45.2 | 59.9 | 36.6 | 26.9 |
| MM | Mean-w/ | 54.6 | 48.6 | 54.6 | 37.6 | 21.3 |
| (2, 3) | Stat-w/o | 59.6 | 45.4 | 59.6 | 37.1 | 27.1 |
| | Stat-w/ (ours) | **61.0** (↑) | 50.0 (↑) | **61.0** (↑) | 40.9 (↑) | 29.7 (↑) |
| | Mean-w/o | 54.6 | 42.2 | 52.5 | 54.5 | 54.6 |
| MM | Mean-w/ | 51.4 | 37.6 | 49.1 | 51.0 | 51.4 |
| (4, 5) | Stat-w/o | 56.4 | 43.7 | 54.4 | **56.3** | **56.4** |
| | Stat-w/ (ours) | 45.1 (↓) | 38.6 (↓) | 43.9 (↓) | 44.3 (↓) | 44.7 (↓) |
| | Mean-w/o | 57.5 | 47.6 | 57.4 | 37.3 | 30.8 |
| MM | Mean-w/ | 53.5 | 49.4 | 53.4 | 37.1 | 23.0 |
| (6, 7) | Stat-w/o | 58.2 | 48.0 | 58.1 | 37.7 | 31.1 |
| | Stat-w/ (ours) | 60.0(↑) | 52.2(↑) | 59.9 (↑) | 41.3 (↑) | 34.4 (↑) |
| | Mean-w/o | 57.6 | 48.1 | 57.5 | 42.8 | 33.6 |
| MM | Mean-w/ | 57.8 | 52.2 | 57.8 | 50.9 | 33.6 |
| (8, 9) | Stat-w/o | 59.7 | 50.1 | 59.5 | 44.2 | 34.6 |
| | Stat-w/ (ours) | 60.4 (↑) | 57.4 (↑) | 60.3 (↑) | 54.3 (↑) | 49.9 (↑) |

Table 8: Accuracies of different models (%) are evaluated on VGGSound when audio features are perturbed. Mean accuracies are reported after repeatedly running 5 times.

| UM / MM | Model | Clean | $\omega_0 = 1.5$ | $\omega_1 = 6$ | $\omega_4 = 0.03$ | $\omega_5 = 0.001$ |
|---|---|---|---|---|---|---|
| | 0: Img-regT | 27.4 | 5.78 | 27.4 | 9.5 | 9.0 |
| | 1: Aud-regT | 54.4 | 15.0 | 49.8 | 23.0 | 19.9 |
| | 2: Img-advT | 27.5 | 5.3 | 27.4 | 10.7 | 10.3 |
| | 3: Aud-advT | 54.8 | 19.4 | 50.6 | 41.8 | 48.5 |
| UM | 4: Img-freT | 22.2 | 4.0 | 22.2 | 19.4 | 20.9 |
| Nets | 5: Aud-freT | 49.2 | 32.4 | 46.6 | 48.2 | 49.3 |
| | 6: Img-staT | 27.0 | 6.9 | 26.9 | 10.5 | 9.6 |
| | 7: Aud-staT | 55.2 | 18.8 | 50.0 | 22.4 | 20.1 |
| | 8: Img-mixT | 27.2 | 8.4 | 27.1 | 7.3 | 7.2 |
| | 9: Aud-mixT | 56.3 | 1.57 | 50.1 | 16.9 | 11.7 |
| | Mean-w/o | 57.7 | 29.5 | 55.5 | 36.2 | 32.7 |
| MM | Mean-w/ | 53.9 | 23.4 | 50.8 | 29.8 | 25.2 |
| (0, 1) | Stat-w/o | 58.5 | 30.5 | 56.1 | 36.6 | 32.9 |
| | Stat-w/ (ours) | **60.0**(↑) | 31.3 (↑) | 52.6 (↓) | 38.9 (↑) | 35.5 (↑) |
| | Mean-w/o | 59.1 | 37.5 | 56.7 | 53.4 | 55.9 |
| MM | Mean-w/ | 54.8 | 29.7 | 52.0 | 43.7 | 48.5 |
| (2, 3) | Stat-w/o | 59.6 | 38.0 | 57.1 | 54.2 | 56.5 |
| | Stat-w/ (ours) | 55.4 (↓) | 38.5 (↑) | 53.2 (↓) | **56.3** (↑) | **57.9** (↑) |
| | Mean-w/o | 54.6 | 39.2 | 54.6 | 54.6 | 54.6 |
| MM | Mean-w/ | 51.4 | **45.4** | 51.3 | 51.0 | 51.5 |
| (4, 5) | Stat-w/o | 56.5 | 39.2 | **58.8** | 55.8 | 55.4 |
| | Stat-w/ (ours) | 46.2 (↓) | 35.5 (↓) | 57.3 (↓) | 55.5 (↓) | 47.3 (↓) |
| | Mean-w/o | 57.5 | 36.2 | 55.7 | 36.7 | 35.2 |
| MM | Mean-w/ | 53.3 | 36.3 | 52.0 | 28.5 | 37.1 |
| (6, 7) | Stat-w/o | 58.2 | 35.7 | 55.0 | 36.9 | 31.0 |
| | Stat-w/ (ours) | 54.4 (↑) | 27.7 (↑) | 50.2 (↑) | 38.8 (↑) | 25.4 (↑) |
| | Mean-w/o | 57.5 | 4.1 | 54.5 | 21.1 | 31.2 |
| MM | Mean-w/ | 57.8 | 3.5 | 53.2 | 17.8 | 26.9 |
| (8, 9) | Stat-w/o | 53.8 | 3.6 | 56.6 | 21.2 | 32.6 |
| | Stat-w/ (ours) | 59.6 (↑) | 5.9 (↑) | 52.7 (↓) | 26.4 (↑) | 33.3 (↑) |

