# OpenReview forum: "Training-Free Robust Multimodal Learning via Sample-Wise Jacobian Regularization"
_ICLR.cc/2022/Conference — ICLR 2022 Submitted_

### Official Review · Reviewer_BBZL · 2021-10-30

**Correctness:** 3
**Technical Novelty And Significance:** 3
**Empirical Novelty And Significance:** 3
**Recommendation:** 6
**Confidence:** 3

**Main Review:**


## Strength

- This paper proposes to utilize Jacobian regularization to improve neural networks' robustness when one of the modality is corrupted. This method is intuitively sound and also provides rigorous guarantee by *Theorem 1* that the corruption can only models' final prediction to a certain amount.
- To find the optimal $W_a$ matrix, this paper also proposes an efficient optimization algorithm.
- In the experiments section, this paper's method could outperform baselines for most of the time.

## Weakness

- This paper makes the assumption that the corrupted modality is known. It would be appreciated if the authors could provide concrete examples about this scenario or references using similar assumption.
- The authors state that their method is *training-free*. Actually, there involves some optimization operation to identify $W_a$. It would be good if the authors explicitly explain what their *trianing-free* means. If possible, some similar definition in other papers would be appreciated. (The authors state that their method is *the first to propose a training-free robust late-fusion method*. It would be important to check whether the authors' definition of *training-free* is consistent with other paper.)
- The rows in *Table 1*, *Table 2*, and *Table 3* are not clear. It would be good if the authors could explicitly define *UM*, *MM*, *(0, k)* mean.

## Questions and minor typos

- In the explanation for *equation (5)*, the authors write *the second term in the loss guarantees numerical stability*. It would be good if the authors could explain what the *numerical stability* means here.
- At the end of Section 4, the authors write *the extra modailty plays a key role*. The *key role* term is vague. Could the authors be explicit about what the role of the extra modality is?
- In the second paragraph, the authors write *our method invoked even larter than (0,2)*. *larter* might be a typo.

- In *equation (4)*, should $P'P^{',T}$ be $P'P^{'T}$ ?

**Summary Of The Paper:**

This paper propses a late-fusion algorithm for multimodal learning. This algorithm serves to improve the robustness against adversarial attacks and random corruptions. Assuming that which modality is perturbed, this paper propses to leverage Jacobian regularization and conditional independence assumption to fuse predictions from different modalities. Moreover, this paper also provides rigorous error bounds on it error and explain the necessity of extra modality.

In summary, this paper's main contributions are as follows:

- Propose a late-fusion algorithm based on Jacobian regularization and conditional independence;
- Derive theoretical error bounds and demonstrate the biasing effect of the extra modality;
- Conduct comprehensive evaluation on adversarial attacks and random corruptions and outperform baseline late-fusion algorithms.

**Summary Of The Review:**


This paper proposes a novel late-fusion algorithm based on Jacobian regularization and conditional independence assumption. It also provides theoretical guarantee on its performance. Further, this paper conducts comprehensive experiments to verify their method. However, there are still some concerns about it assumption and claims.

---

> ### Author Response · Authors · 2021-11-10
> **Response to Reviewer BBZL (Part I)**
>
> Thanks for the constructive comments. The authors have made changes correspondingly in the paper. In what follows, we reply to each of the reviewer's concerns.
>
> - **Weakness #1**:
>
> This paper makes the assumption that the corrupted modality is known. It would be appreciated if the authors could provide concrete examples about this scenario or references using similar assumption.
>
> - **Response**:
>
> The authors first want to provide a concrete example of this scenario. In a self-driving vehicle, RGB cameras and LiDARs are usually deployed simultaneously and the resulting RGB images and point clouds are referred to as two modalities, respectively. Moreover, it is known that RGB images are blurry at night corrupted by noises at pixels, but point clouds work well. Consider now we construct a dataset including all RGB images and point clouds collected at night. In this dataset, the RGB image is the corrupted modality but not the point cloud. Second, the authors would like to point out the previous work [1] on robust (middle-level) multimodal learning, has made this assumption.
>
> Moreover, the authors emphasize that we don't assume the corrupted modality is known. In the section of the proposed method, we say this mainly for omitting the calculation of $W_b$ (i.e., for ease of writing). But in our numerical experiments, we have considered all different settings. To address the reviewer's concern, the tables of particular interests might be (i) Table 5: Both modalities are perturbed by Gaussian noises on AV-MNIST, and we calculate both $\{W_a, W_b\}$, (ii) Table 7: the video features are perturbed on VGGSound. But we do not know which modalities are perturbed, so we invoke robust add-ons for both modalities (i.e., calculating both $\{W_a, W_b\}$), (iii) Table 8: the audio features are perturbed on VGGSound. But we do not know which modalities are perturbed, so we invoke robust add-ons for both modalities (i.e., calculating both $\{W_a, W_b\}$).
>
> Last but not least, in our initial submission, we have already said in Appendix B.4 that ‘... because in real-world applications, we sometimes do not know which modality is corrupted.' We apologize that we didn't make this explicit in the main text and caused the confusion. The authors have made clarifications in both the Method section and the Experiment section of the revised main text.
>
> - **Weakness #2**:
>
> The authors state that their method is training-free. Actually, there involves some optimization operation to identify $W_a$. It would be good if the authors explicitly explain what their training-free means. If possible, some similar definition in other papers would be appreciated. (The authors state that their method is the first to propose a training-free robust late-fusion method. It would be important to check whether the authors' definition of training-free is consistent with other paper.)
>
> - **Response**:
>
> By saying we are the first to propose a training-free robust late-fusion method, our emphasis is on ‘robust late-fusion'. This is rather unexplored in the literature: we are the first on robust multimodal late-level fusion. Here 'training-free’ refers to the fact that (a) we do not need to re-train the unimodal/multimodal networks and (b) only introduce minor computational overhead in the inference stage.
>
> To be specific, (a) the most prominent approach to defend against adversarial attacks, adversarial training, requires retraining multimodal networks. This is resource-intensive and may not be practical in real-life scenarios. For instance, for the hospital scenario in the introduction section, retraining a model is not possible due to data privacy while our method can directly utilize the unimodal networks available at hand to improve robustness;
>
> (b) We introduce few computations in the inference stage. As the reviewer mentioned, optimization is still needed. However, as demonstrated in the Appendix, the optimization problem is relaxed to a Sylvester equation as shown in Eq (8) and it has an analytical solution. In other words, we don't need to actually solve it by an optimization solver, but rather directly calculate the solution by formula. Namely, in Step 4 of Algorithm 1, we do a forward pass to calculate the Jacobian matrix, where the forward pass is needed anyway in the inference time of a neural network. In Step 5, we use the analytical solution of the Sylvester equation. Finally, in step 6, we use the calculated $W_a$ and Eq (3) to obtain the final prediction. There is no time-consuming training or optimization needed to be done.
>
> Last but not least, we really appreciate the reviewer's thoughtful comments on the term ‘training-free'. If the reviewer believes it is aggressive and over-claiming, the authors will delete it as required. We would really appreciate it if the reviewer could provide any further suggestions.
>
> **---------- Due to the characters limit, the remaining response is in Part II. ----------**

---

> > ### Author Response · Authors · 2021-11-10
> > **Response to Reviewer BBZL (Part II)**
> >
> > **---------- Response continued here ----------**
> >
> >
> > - **Weakness #3**:
> >
> > The rows in Table 1, Table 2, and Table 3 are not clear. It would be good if the authors could explicitly define UM, MM, (0, k) mean.
> >
> > - **Response**:
> >
> > ‘UM' and ‘MM' are short for ‘unimodal' and ‘multimodal', respectively. As of (0,k), it means the unimodal backbone. For instance, the row ‘MM(0,1)' means a multimodal network obtained by fusing the unimodal network indexed by 0 and 1. We have added explanations in the caption of Table 1 in the revised paper.
> >
> > - **Question #1**:
> >
> > In the explanation for equation (5), the authors write the second term in the loss guarantees numerical stability. It would be good if the authors could explain what the numerical stability means here.
> >
> > - **Response**:
> >
> > In a nutshell, when $\gamma$ equals $0$ (corresponding to no second term in Eq (5)), the Sylvester equation is ill-conditioned. To be more specific, as shown in Eq (8), when $\gamma\to 0$, the matrix $A$ will tend to infinity, leading to an ill-conditioned matrix equation. We have added explanations after Eq (8) in the revised paper to make this clear.
> >
> > - **Question #2**:
> >
> > At the end of Section 4, the authors write the extra modality plays a key role. The key role term is vague. Could the authors be explicit about what the role of the extra modality is?
> >
> > - **Response**:
> >
> > The extra modality exerts a biasing effect as explained in Section 4. The intuition behind is that unimodality plus Jacobian regularization doesn’t work as there is no ‘reference' model to adjust the model's prediction. We do not know in which direction should we adjust the model for better robustness. On the contrary, in the multimodal context, multiple modalities serve as ‘reference' models for each other and point a direction. This corresponds to the $z_B$ term in the last paragraph of Section 4. Besides the formula provided, we use the TwoMoon example for easy illustration and hope these two can shed light on the effect of the extra modality.
> >
> > - **Question #3**:
> >
> >  In the second paragraph, the authors write our method invoked even larter than (0,2). larter might be a typo.
> >
> > - **Response**:
> >
> > ‘larter’ is a typo. It should be ‘larger’. We have corrected it in the revised paper.
> >
> > - **Question #4**:
> >
> > In equation (4), should $PP^{\prime,T}$ be $PP^{\prime\,T}$ ?
> >
> > - **Response**:
> >
> > The authors use $P^{\prime,T}$ to represent the transpose of $P^\prime$. To make it clear, the authors are used to adding a comma between the prime and transpose. We were wondering does omitting the comma appear to be much clearer? If so, we could remove the comma in our next version.
> >
> > **Authors’ Note**
> >
> > In summary, the authors highly appreciate the effort and time of the reviewer. We hope our explanations have addressed the reviewer's concerns. We are open to any more discussions. If there are no other strong opinions on the paper, the authors would really appreciate it if the reviewer could consider raising the score. :)
> >
> > **Reference**
> >
> > [1] Taewan Kim and Joydeep Ghosh, ‘On Single Source Robustness in Deep Fusion Models', NeurIPS 2019.

---

### Official Review · Reviewer_dv5k · 2021-10-31

**Correctness:** 4
**Technical Novelty And Significance:** 3
**Empirical Novelty And Significance:** 3
**Recommendation:** 5
**Confidence:** 4

**Main Review:**

Strengths:

- The idea is interesting and well-motivated
- The empirical results on various datasets are solid

Weakness/major concerns:

- To make eq(1) more rigorous, I would suggest to note the shape of $W_{A/B}$ and $h_A$.
- in Eq(2) $freq$ contains the occuring frequencies of each class, calculated from the train dataset. Will it be directly applied to compute $p$ on test dataset? Looks like this approach only invoked at test time, so I think the answer is yes. Here $freq$ denotes the class prior which is crucial for computing $p$ accurately. I think when domain shift, more specifically label shift exsit between train and test dataset, it is inappropriate to estimate the test class prior from train dataset. Does the proposed method can handle this problem?
- In eq(5), is the objective $\min_{W_a} L$ or $\min_{W_a,W_A} L$? According to the Algorithm 1, it minimizes over $W_a$. It would be better to clarify in the equation as well.
- To me the error bound in Theorem 1 is very loose. Is it possible to compute the value exactly for some experiment settings, for instance biase noise? We can make some moderate assumptions, like set the Lipschitz constant to 1.
- Is this the work pipline of the algorithm, at training time, we fuse $z_A$ and $z_B$ directly; at test time, we fuse $W_a z_A$ and $W_b z_B$ with optimized $W_a$? Do we have to solve the Sylvester equation for each mini-batch? I would appreicate if the authors can help me to get a better understanding  of their algorithm.
- For AV-MNIST experiment, would it be more interesting to perturb the image modality? Intuitively, image modal provide more useful information than audio modality. But of course this is not necessary to be true.

**Summary Of The Paper:**

In this paper, the authors proposed a training-free late-fusion method for robust multimodal learning. They specifically considering its performance under adversarial attacks and random corruptions which usually confuse the model by introducing noise to the input data. To promote the multimodal prediction robust to attacks, they propose to minimize the Frobenius norm of Jacobian matrix so that the prediction becomes stable to the perturbation of inputs. They also provide a theoretical error bound of their method. The experimental results outperfom other late-fusion methods.

**Summary Of The Review:**

I would consider to raise my score, if the authors can address my concern.

---

> ### Author Response · Authors · 2021-11-10
> **Response to Reviewer dv5k (Part I)**
>
> Thanks for the constructive comments. The authors have made changes correspondingly in the paper. In what follows, we reply to each of the reviewer's concerns.
>
> - **Weakness #1**:
>
> To make eq(1) more rigorous, I would suggest to note the shape of $W_{A/B}$ and $h_A$.
>
> - **Response**:
>
> We have explicitly written out the dimension of $\{W_A,h_A,b_A\}$ in the paragraph below Eq(1) of the revised paper.
>
> - **Weakness #2**:
>
> In Eq(2) $freq$ contains the occuring frequencies of each class, calculated from the train dataset. Will it be directly applied to compute $p$ on test dataset? Looks like this approach only invoked at test time, so I think the answer is yes. Here $freq$ denotes the class prior which is crucial for computing $p$ accurately. I think when domain shift, more specifically label shift exsit between train and test dataset, it is inappropriate to estimate the test class prior from train dataset. Does the proposed method can handle this problem?
>
> - **Response**:
>
> The authors completely agree with the reviewer. The $freq$ essentially represents the prior probability of each class. Considering that we only have true labels for samples in the train dataset, the best we could do is to calculate $freq$ on train dataset and reuse it for test data. This implicitly assumes that the train data and test data are drawn from the same distribution, which is a common assumption made in most ML applications.
>
> On the other hand, when the train and test data come from different distributions, referred to as the out-of-domain (OOD) problem by many, we honestly admit that our present method doesn't work due to domain shift. However, a simple yet efficient remedy might be that we make $freq$ a learnable parameter and initialize it with the value calculated on the train dataset. We will test our method in an OOD setting in the future. This discussion has been added at the end of the Conclusion section of the revised paper.
>
> - **Weakness #3**:
>
> In eq(5), is the objective $min_{W_a}L$ or $min_{W_A}L$? According to Algorithm 1, it minimizes over $W_a$. It would be better to clarify the equation as well.
>
> - **Response**:
>
> The optimization is solved solely with respect to $W_a$. Our intuitive idea is that when a modality is perturbed either by adversarial attacks or random corruptions, we compensate this perturbation by a carefully designed $W_a$ inserted right before the late fusion. $W_A$ is the fixed weight parameter. We have revised Eq (5) to make this explicit.
>
> - **Weakness #4**:
>
> To me the error bound in Theorem 1 is very loose. Is it possible to compute the value exactly for some experiment settings, for instance biase noise? We can make some moderate assumptions, like set the Lipschitz constant to 1.
>
> - **Response**:
>
> The authors agree that the bound might not be tight. Please bear with us as here our goal is not to obtain a tightest bound, but primarily demonstrate that by using our method, the solution has some kind of guarantees. As mentioned by the reviewer, in some cases such as bias noise and Gaussian noise, $\epsilon$ could be quantified and thus we could directly obtain a real value of the bound. For instance, (i) when ${\epsilon}\sim N(\mathbf{0},\Sigma)$, the bound is simplified to  ${E}[||\mathbf{p}^{\prime,noise}-\mathbf{p}^\prime||]\leq l({\frac{\gamma K}{2(1-\gamma)}})^{1/2}\text{Tr}[\mathbf{\Sigma}]$, (ii) when the $L_2$ norm of ${\epsilon}$ is constrained smaller than $\delta$ (usually assumed in adversarial attacks), the bound is simplified to $||\mathbf{p}^{\prime,noise}-\mathbf{p}^\prime||\leq l\delta({\frac{\gamma K}{2(1-\gamma)}})^{1/2}$. These could be treated as the corollary of Theorem 1, and have been added in the paragraph before Theorem 1 of the revised paper.
>
> **---------- Due to the characters limit, the remaining response is in Part II. ----------**

---

> > ### Author Response · Authors · 2021-11-10
> > **Response to Reviewer dv5k (Part II)**
> >
> > **---------- Response continued here ----------**
> >
> >
> > - **Weakness #5**:
> >
> > Is this the work pipline of the algorithm, at training time, we fuse $z_A$ and $z_B$ directly; at test time, we fuse $W_az_A$ and $W_bz_B$ with optimized ? Do we have to solve the Sylvester equation for each mini-batch? I would appreicate if the authors can help me to get a better understanding of their algorithm.
> >
> > - **Response**:
> >
> > Based on the reviewer's description, the authors believe that the reviewer has understood our method correctly. Here to explain in more detail, if the multimodal train data are provided not in pairs, as in the medical application mentioned in the Introduction section, each unimodal network is trained separately. Afterward, the multimodal network is obtained by Eq (2). On the other hand, if the multimodal train data are presented in pair $(x_A, x_B, y)$, besides the above training method, an optional one is that we could directly fuse $z_A$ and $z_B$ and train a multimodal network.
> >
> > No matter which way, in inference time (i.e., on test dataset), we calculate $W_a$ and $W_b$ for each input sample according to Algorithm 1, and the final multimodal output is given by fusing $z_A'=W_az_A$ and $z_B'=W_bz_B$ according to Eq (3). Intuitively, we are designing a ‘filter' {$\{W_a,W_b\}$} for every sample to compensate for the perturbation on it. Since the optimization involved to solve {$\{W_a,W_b\}$} has been relaxed to a Sylvester equation, it only leads to marginal computation overhead.
> >
> > - **Weakness #6**:
> >
> > For AV-MNIST experiment, would it be more interesting to perturb the image modality? Intuitively, image modal provide more useful information than audio modality. But of course this is not necessary to be true.
> >
> > - **Response**:
> >
> > In the appendix of our initial submission, numerical results on perturbing image modality and perturbing both modalities are already included. Moreover, this is also true for the RAVDESS and VGGSound experiments. We apologize that we haven't made this clear in the main text. We have added sentences in the main text to emphasize that the results of other settings have been included in Appendix, e.g., the last sentence right above Figure 4.
> >
> > **Authors' Note**
> >
> > In summary, the authors highly appreciate the effort and time of the reviewer. We hope our explanations have addressed the reviewer's concerns. We are open to any more discussions!

---

> > > ### Comment · Reviewer_dv5k · 2021-12-06
> > > **Thanks the author's responses!**
> > >
> > > - Weakness #2
> > >
> > > "a simple yet efficient remedy might be that we make $freq$ a learnable parameter and initialize it with the value calculated on the train dataset"
> > >
> > > I doubt this gonna be an **efficient** remedy. You can't expect to learn a $freq$, that can adapt well to test data, from the training dataset. At least, the model should be trained in a transductive manner.
> > >
> > > - Weakness #4
> > >
> > > "Please bear with us as here our goal is not to obtain a tightest bound, but primarily demonstrate that by using our method, the solution has some kind of guarantees"
> > >
> > > When your bound is loose, it guarantees nothing. Since the empirical results look good, I am not sure whether it would be better to not emphasize the general bound. Perhaps you can move it to appendix.
> > >
> > > Overall, I think the paper was well motivated. But I would suggest to further polish.

---

> > > > ### Author Response · Authors · 2021-12-06
> > > > **Thanks for the reviewer's responses!**
> > > >
> > > > * Reply to Weakness #2
> > > >
> > > > Thanks for the comments. The authors would like to politely point out that the consideration of out-of-domain setting (or domain adaptation) is itself an open/hot topic in the research literature. We would consider adapting our method to suit this setting in the future, but it appears to be another new project.
> > > >
> > > > * Reply we Weakness #4
> > > >
> > > > Thanks for the comments. The authors will move the theoretical results to the Appendix. On the other hand, the authors would really appreciate it if the reviewer could provide the reasons why he/she thought the bound is loose and give some specific hints about how to further improve it.
> > > >
> > > > If there are no strong opinions, the authors would really appreciate it if the reviewer could consider raising the score as he/she mentioned previously.

---

> ### Comment · Area_Chair_iD5h · 2021-12-04
> **Please read and respond to the authors' rebuttal**
>
> Though this is late already, please still try to respond to the authors' rebuttal as soon as possible. Thanks!
>
> AC

---

### Official Review · Reviewer_oUrr · 2021-11-03

**Correctness:** 3
**Technical Novelty And Significance:** 2
**Empirical Novelty And Significance:** 2
**Recommendation:** 5
**Confidence:** 3

**Main Review:**

Strengths:

- The Jacobian regularization methods is simple and straight forward.
- The paper provide a theoretical error bound of the proposed robust late-fusion method.
- The ablation of \gamma on the necessity of extra modalities via the TwoMoon example is interesting.

Weaknesses:

- In the related work section, the paper claim (robust) late-fusion in multimodal learning is un-explored in the previous literature, However, this is not entirely true, some of the missing references includes [1][2].
- The paper provide some experimental results, i.e. Multimodal classfication dataset AV-MNIST/VGGSound and Emotion Recogition dataset RAVDESS. However, this is not a extensive in multimodal learning. More real-world tasks and datasets should be considered, e.g. RGB-D Human Action Recognition dataset NTU-RGBD, multimodal classfication dataset Kinetics400/AudioSet.

More comments:

- In their implementation, the author mentioned one iteration could already yield a sufficient accurate result (tmax = 1), I'm a little bit suprise on this, can you elaborate more on this?


[1] MMTM: Multimodal Transfer Module for CNN Fusion, CVPR 2020
[2] Deep Multimodal Fusion by Channel Exchanging, NeurIPS

**Summary Of The Paper:**

The paper proposed a training-free robust multimodal learning late-fusion methods via sample-wise Jacobian regularization. The key idea of the work is to minimize the Frobenius norm of a Jacobian matrix, so that the multimodal prediction is stabilized. The paper demonstrate
the good efficacy on both adversarial attacks and random corruptions setting on multimoda datasets such as AV-MNIST/VGGSound/RAVDESS.


**Summary Of The Review:**

The paper proposed a simple multimodal fusion methods based on Jacobian regularization, However, there are minor issue in related work, and experiments are not extensive, i.e. only valided on smaller Toy datasets.

---

> ### Author Response · Authors · 2021-11-10
> **Response to Reviewer oUrr**
>
> Thanks for the thoughtful comments. The authors do value the efforts and time of the reviewer, but we respectively argue that the mentioned weakness points do not make sense.
>
> - **Weakness #1**:
>
> In the related work section, the paper claim (robust) late-fusion in multimodal learning is un-explored in the previous literature, However, this is not entirely true, some of the missing references includes [1][2].
>
> - **Response**:
>
> The authors believe this work is the first on robust late-fusion in the context of multimodal learning. Both the mentioned papers are about middle-fusion (not late-fusion), which are explicitly demonstrated in both of their Figure 1 (c) at the beginning. Also, there isn't any word on robustness in these papers, nor there are any experiments conducted with adversarial attacks or random corruptions. With all respect, the authors do not see why these two papers are highly related. The authors would really appreciate it if the reviewer could further explain his/her comments. If there are other papers on robust late-level multimodal fusion, we are happy to revise our claim, include and compare our method with them.
>
> - **Weakness #2**:
>
> The paper provide some experimental results, i.e. Multimodal classification dataset AV-MNIST/VGGSound and Emotion Recognition dataset RAVDESS. However, this is not a extensive in multimodal learning. More real-world tasks and datasets should be considered, e.g. RGB-D Human Action Recognition dataset NTU-RGBD, multimodal classification dataset Kinetics400/AudioSet.
>
> - **Response**:
>
> The authors strongly disagree that VGGSound is termed a toy dataset by the reviewer. VGGSound contains over 200k clips for 300 different sound classes [1]. VGGSound is a well-known large-scale benchmark in multimodal classification, extensively used by many papers, such as [1,2,3,4,5,6]. When they describe VGGSound, ‘large-scale' occurs frequently. To demonstrate a bit more, in the experiment section of [2], they mentioned that ‘... only VGG-Sound and Kinetics400 are large enough for learning strong representations from scratch'. It implies that VGGSound is considered comparable to Kinetics400 in the ML community.
>
> While the authors agree that NTU-RGBD is a good benchmark, it is impossible to run experiments on every public dataset on the Internet. Actually, the authors have already carefully selected the datasets to cover all sizes: (i) small: TwoMoon, AV-MNIST, (ii) medium: RAVDESS, and (iii) large: VGGSound. Also, in the appendix, a large number of ablations have been done, and the authors believe the current numerical results are sufficient to verify the efficacy of the proposed method. Again, the authors are open to any more comments.
>
> - **Remark #1**:
>
> In their implementation, the author mentioned one iteration could already yield a sufficient accurate result (tmax = 1), I'm a little bit suprise on this, can you elaborate more on this?
>
> - **Response**:
>
> Choosing tmax=1 involves two-fold consideration. First, in our numerical results, we witness that choosing tmax=1 could already bring satisfying improvement. Second, we need to solve a matrix Wa for every sample using an iterative process shown in Algorithm 1. For each sample and in each iteration, it requires one forward propagation (in Step 4 shown in Algorithm 1), resulting in N*Tmax forward passes in total for a test dataset with size of N. In other words, choosing tmax=1 is a balance between performance and run time. This strategy is commonly used in many iterative methods. To name one, in [7], it exploits a Bayesian expectation maximization (EM) framework, and it sets the number of iteration T=2.
>
>
> **Authors' Note**
>
> With all respects, the authors believe that none of the two weak points hold. To this end, we would really appreciate it if the reviewer could reconsider his/her comments and score. The authors are happy to discuss any additional remarks the reviewer will make.
>
> **Reference**
>
> [1] Honglie Che, et al., ‘VGGSound: A Large-scale Audio-Visual Dataset‘, ArXiv 2021.
>
> [2] Yuki M. Asan et al., ‘Labelling unlabelled videos from scratch with multi-modal self-supervision', NeurIPS 2021.
>
> [3] Zihui Xue, et al., 'Multimodal Knowledge Expansion', ICCV 2021.
>
> [4] Yanbei Chen et al., ‘Distilling Audio-Visual Knowledge by Compositional Contrastive Learning', CVPR 2021.
>
> [5] Xuhui Jia et al., ‘Joint Representation Learning and Novel Category Discovery on Single- and Multi-modal Data', ICCV 2021.
>
> [6] Fonseca, Eduardo et al., ‘FSD50k: an open dataset of human-labeled sound events', ArXiv 2020.
>
> [7] Ashish Khetan, et al., ‘Learning from noisy singly-labeled data', ICLR 2018.

---

> ### Comment · Area_Chair_iD5h · 2021-12-04
> **Please read and respond to the authors' rebuttal**
>
> Dear oUrr,
>
> Though this is late already, please try to read and respond to the authors' rebuttal as soon as possible. Thanks!
>
> AC

---

### Decision · Program_Chairs · 2022-01-20

**Decision:**

Reject

**Comment:**

Three experts reviewed the paper and gave mixed reviews. Reviewer BBZL raised their score to 6 in the discussion phase. Reviewer dv5k was not fully convinced by the rebuttal and remained negative. Reviewer oUrr also remained negative. The reviewers were not excited by the proposed method in general and raised questions about both experiments and theoretical results. AC found clear merits in the paper, but the reviewers' comments suggested the work could be strengthened in both experiments and presentation. Hence, the decision is *not* to recommend acceptance at this time. The authors are encouraged to consider the reviewers' comments when revising the paper for submission elsewhere.